# DISTRIBUTED LEAST SQUARE RANKING WITH RANDOM FEATURES

## ABSTRACT

In this paper, we study the statistical properties of pairwise ranking using distributed learning and random features (called DRank-RF) and establish its convergence analysis in probability. Theoretical analysis shows that DRank-RF remarkably reduces the computational requirements while preserving a satisfactory convergence rate. An extensive experiment verifies the effectiveness of DRank-RF. Furthermore, to improve the learning performance of DRank-RF, we propose an effective communication strategy for it and demonstrate the power of communications via theoretical assessments and numerical experiments.

## 1 INTRODUCTION

Distributed learning has attracted much attention in the literature and has been widely used for kernel learning in large scale scenarios (Zhang et al., 2013; Chang et al., 2017; Lin et al., 2020b). The distributed kernel learning has mainly three ingredients: Processing the data subset in the local kernel machines and producing a local estimator; Communicating exclusive information such as the data (Bellet et al., 2015), gradients (Zeng & Yin, 2018) and local estimator (Huang & Huo, 2019) between the local processors and the global processor; Synthesizing the local estimators and the communicated information on the global processor to produce a global estimator. Note that, in the divide-and-conquer learning, the second ingredient communications is not necessary. In the terms of practical challenges and theoretical analysis, the distributed learning has made significant breakthroughs in the multi-penalty regularization (Guo et al., 2019), coefficient-based regularization (Pang & Sun, 2018), spectral algorithms (Mücke & Blanchard, 2018; Lin et al., 2020a), kernel ridgel regression (Yin et al., 2020; 2021), and semi-supervised regression (Li et al., 2022). All the above are restricted to pointwise kernel learning. However, the distributed learning in pairwise kernel learning still has a long way to go. The existing distributed pairwise learning (Chen et al., 2019; 2021) has high computational requirements, which motivates us to explore theoretic foundations and efficient methods for pairwise ranking kernel methods under distributed learning.

Random features methods (Rahimi & Recht, 2007; Carratino et al., 2018; Liu et al., 2021) have a long and distinguished history, which embed the non-linear feature space (i.e. the Reproducing Kernel Hilbert Space associated with the kernel) into a low dimensional Euclidean space while incurring an arbitrarily small additive distortion in the inner product values. This enables one to overcome the high computational requirements of kernel learning since one can now work in an explicit low dimensional space with explicit representation whose complexity depends only on the dimensionality of the space. Random features have gained rapid progress in reducing the complexity of the kernel ridge regression (Liu et al., 2021) and semi-supervised regression (Li et al., 2022).However, it remains unclear for complexity reduction and learning theory analysis to the distributed pairwise ranking kernel learning.

In this paper, to reduce the computational requirements of pairwise ranking kernel learning, we investigate the method of combining distributed learning and random features for pairwise ranking kernel learning, called distributed least square ranking with random features (DRank-RF), to deal with large-scale applications, and study its statistical properties in probability by integral operators framework. To further improve the performance of DRank-RF, we consider communications among different local processors. The main contributions of this paper are as follows: 1) We construct a novel method DRank-RF to improve the existing state-of-the-art performance of the distributed pairwise ranking kernel learning. This work is the first time to apply random features to least square

ranking and derive the theoretical guarantees, which is a new exploration of random features in least square ranking. In theoretical analysis, we derive the convergence rate of the proposed method, which is sharper than that of the existing state-of-the-art distributed pairwise ranking kernel learning (See Theorem 1). In computational complexity, DRank-RF requires essentially $\mathcal{O}(m^2|D_j|)$ time and $\mathcal{O}(m|D_j|)$ memory, where $m$ is the number of random features, $m < |D_j|$, and $|D_j|$ is the number of data in each local processor. The proposed method can greatly reduce the computational requirements compared with the state-of-the-art works (See Table 1). Experimental results verify that the proposed method keeps the similar testing error as the exact and state-of-the-art approximate kernel least square ranking and has a great advantage over the exact and state-of-the-art approximate kernel least square ranking in the training time, which is consistent with our theoretical analysis. 2) We propose a communication strategy to further improve the performance of DRank-RF, called DRank-RF-C. Statistical analysis shows that DRank-RF-C obtains a faster convergence rate with the help of communication strategy than DRank-RF. And the numerical results validate the power of the proposed communication strategy.

The paper is organized as follows: In section 2, we briefly introduce the least square ranking problem and the distributed least square ranking. In section 3, we introduce the proposed methods. Section 4 shows the theoretical analysis of the proposed DRank-RF and DRank-RF-C. In section 5, we compare the related works with the proposed methods. The following sections are the experiments and conclusions.

## 2 BACKGROUND

There is a compact metric space $\mathcal{Z} := (\mathcal{X}, \mathcal{Y}) \subset \mathbb{R}^{q+1}$, where $\mathcal{X} \subset \mathbb{R}^q$ and $\mathcal{Y} \subset [-b, b]$ for some positive constant $b$. The sample set $D := \{(\mathbf{x}_i, y_i)\}_{i=1}^N$ of size $N = |D|$ is drawn independently from an intrinsic Borel probability measure $\rho$ on $\mathcal{Z}$. $\rho(y|X = \mathbf{x})$ denotes the conditional distribution for given input $\mathbf{x}$. The hypothesis space used is the reproducing kernel Hilbert space (RKHS) ($\mathcal{H}_K$) associated with a mercer kernel $K : \mathcal{X} \times \mathcal{X} \to \mathbb{R}$ (Aronszajn, 1950). We will denote the inner product in $\mathcal{H}_K$ by $\langle \cdot, \cdot \rangle$, and corresponding norm by $\| \cdot \|_K$.

### 2.1 LEAST SQUARE RANKING (LSRANK)

Least square ranking (LSRank) is one of the most popular learning methods in the machine learning community (Chen, 2012; Zhao et al., 2017; Chen et al., 2019),which can be stated as $f_{D,\lambda} = \arg\min_{f \in \mathcal{H}_K} \left\{ \mathcal{E}_D(f) + \lambda\|f\|_K^2 \right\}$ and $\mathcal{E}_D(f) = \frac{1}{|D|^2} \sum_{i,k=1}^{|D|} (y_i - y_k - (f(\mathbf{x}_i) - f(\mathbf{x}_k)))^2$, where the regularized parameter $\lambda > 0$.

The main purpose of LSRank is to find a function $f : \mathcal{X} \to \mathbb{R}$ through empirical observation, so that the ranking risk

$$\mathcal{E}(f) = \int_{\mathcal{Z}} \int_{\mathcal{Z}} (y - y' - (f(\mathbf{x}) - f(\mathbf{x}')))^2 \, d\rho(\mathbf{x}, y) d\rho(\mathbf{x}', y') \tag{1}$$

can be as small as possible, where $\mathbf{x}, \mathbf{x}' \in \mathcal{X}$.

The optimal predictor (Chen, 2012; Chen et al., 2013; Kriukova et al., 2016) under Eq.(1) is the regression function $f_\rho(\mathbf{x}) = \int_{\mathcal{Y}} y d\rho(y|X = \mathbf{x}), \mathbf{x} \in \mathcal{X}$.

**Complexity Analysis** LSRank requires $\mathcal{O}(|D|^3)$ time and $\mathcal{O}(|D|^2)$ space, which is prohibitive for large-scale settings.

### 2.2 DISTRIBUTED LEAST SQUARE RANKING (DRANK)

Let the dataset $D = \cup_{j=1}^p D_j$ and each subset $D_j := \left\{ \left( \mathbf{x}_i^j, y_j^j \right) \right\}_{i=1}^{|D_j|}$ be stored in the $j$-th local processor for $1 \le j \le p$. The DRank is defined by

$$\bar{f}_{D,\lambda}^0 = \sum_{j=1}^p \frac{|D_j|^2}{\sum_{k=1}^p |D_k|^2} f_{D_j, \lambda} \tag{2}$$

where the least squares ranking (LSRank) $f_{D_j,\lambda} = \arg\min_{f \in \mathcal{H}_K} \{\mathcal{E}_{D_j}(f) + \lambda \|f\|_K^2\}$ and $\mathcal{E}_{D_j}(f) = \frac{1}{|D_j|^2} \sum_{i,k=1}^{|D_j|} \left( y_i^j - y_k^j - \left( f\left(\mathbf{x}_i^j\right) - f\left(\mathbf{x}_k^j\right) \right) \right)^2$.

**Complexity Analysis** The time complexity, space complexity, and communication complexity of DRank for each local processor are $\mathcal{O}(|D_j|^3)$, $\mathcal{O}(|D_j|^2)$, and $\mathcal{O}(|D_j|)$, where $j = 1, \ldots, p$ and $p$ is the number of partitions.

## 3 PROPOSED ALGORITHMS

### 3.1 DISTRIBUTED LEAST SQUARE RANKING WITH RANDOM FEATURES (DRANK-RF)

Here we first introduce the main properties of the shift-invariant kernel and the basic idea of random features. The shift-invariant kernel can be written as $K(\mathbf{x}, \mathbf{x}') = \int_\Omega \psi(\mathbf{x}, \boldsymbol{\omega}) \psi(\mathbf{x}', \boldsymbol{\omega}) \varrho(\boldsymbol{\omega}) d\boldsymbol{\omega}$ if the spectral measure has a density function $\varrho(\cdot)$ (Li et al., 2019; Carratino et al., 2018), where $\psi : \mathcal{X} \times \Omega \to \mathbb{R}$ is a bounded and continuous function with respect to $\boldsymbol{\omega}$ and $\mathbf{x}$. The basic idea of random features is to approximate the kernel function $K(\mathbf{x}, \mathbf{x}')$ by its Monte-Carlo estimation (Li et al., 2019; Rahimi & Recht, 2007): $K_m(\mathbf{x}, \mathbf{x}') = \frac{1}{m} \sum_{i=1}^m \psi(\mathbf{x}, \boldsymbol{\omega}_i) \psi(\mathbf{x}', \boldsymbol{\omega}_i) = \langle \phi_m(\mathbf{x}), \phi_m(\mathbf{x}') \rangle$, where $\phi_m(\mathbf{x}) = \frac{1}{\sqrt{m}} (\psi(\mathbf{x}, \boldsymbol{\omega}_1), \ldots, \psi(\mathbf{x}, \boldsymbol{\omega}_m))^T$.

Back to supervised learning (Chen, 2012), combining random features with the least squares ranking leads to, $f_{m,D,\lambda}(\mathbf{x}) = \mathbf{g}_{m,D,\lambda}^T \phi_m(\mathbf{x})$ with

$$\mathbf{g}_{m,D,\lambda} = (\boldsymbol{\Phi}_{m,D} \mathbf{W}_D \boldsymbol{\Phi}_{m,D}^T + \frac{\lambda}{2}\mathbf{I})^{-1} \boldsymbol{\Phi}_{m,D} \mathbf{W}_D \bar{\mathbf{y}}_D, \tag{3}$$

for $\boldsymbol{\Phi}_{m,D} = \frac{1}{\sqrt{|D|}}(\phi_m(\mathbf{x}_1), \ldots, \phi_m(\mathbf{x}_{|D|}))$, $\mathbf{W}_D = \mathbf{I}_{|D|} - \frac{1}{|D|}\mathbf{1}_{|D|}\mathbf{1}_{|D|}^T = \frac{1}{|D|}(|D|\mathbf{I} - \mathbf{1}_{|D|}\mathbf{1}_{|D|}^T)$, the identity matrix $\mathbf{I}_{|D|}$, $\mathbf{1}_{|D|} = (1, \ldots, 1)^T \in \mathbb{R}^{|D|}$, and $\bar{\mathbf{y}}_D = \frac{1}{\sqrt{|D|}}(y_1, \ldots, y_{|D|})^T$.

DRank with random features (DRank-RF) is defined as

$$\bar{f}_{m,D,\lambda}^0 = \sum_{j=1}^p \frac{|D_j|^2}{\sum_{k=1}^p |D_k|^2} f_{m,D_j,\lambda}, \tag{4}$$

where $f_{m,D_j,\lambda} = \mathbf{g}_{m,D_j,\lambda}^T \phi_m(\mathbf{x})$ with $\mathbf{g}_{m,D_j,\lambda} = (\boldsymbol{\Phi}_{m,D_j} \mathbf{W}_{D_j} \boldsymbol{\Phi}_{m,D_j}^T + \frac{\lambda}{2}\mathbf{I})^{-1} \boldsymbol{\Phi}_{m,D_j} \mathbf{W}_{D_j} \bar{\mathbf{y}}_{D_j}$.

Random features have a long history and have been studied in different learning, for example kernel ridge regression (Liu et al., 2021), kernel classification (Liu et al., 2022), kernel k-means (Chitta et al., 2012). However, random features have not been studied in least square ranking. Our work is the first time to apply random features to least square ranking and derive the theoretical guarantees, which is a new exploration of the application of random features. In addition, due to the different objective functions and integral operators, the proof of our proposed method is different from the existing methods (See Appendix). Finally, the proposed methods greatly reduce the computational requirements (See Table 1).

The method of synthesis operation in Eq.(4) is to weighted average the estimated values in each local processor.

**Complexity Analysis** In time complexity, solving the inverse of $\boldsymbol{\Phi}_{m,D_j} \mathbf{W}_{D_j} \boldsymbol{\Phi}_{m,D_j}^T + \frac{\lambda}{2}\mathbf{I}$ needs $\mathcal{O}(m^3)$ time and computing the matrices multiplication $\boldsymbol{\Phi}_{m,D_j} \mathbf{W}_{D_j} \boldsymbol{\Phi}_{m,D_j}^T$ requires $\mathcal{O}(m^2|D_j|)$ cost, where $m$ is the number of random features. In space complexity, the key is to store $\boldsymbol{\Phi}_{m,D_j}$, whose space complexity is $\mathcal{O}(m|D_j|)$. Therefore, the time complexity, space complexity, and communication complexity of DRank-RF for each local processor are $\mathcal{O}(m^2|D_j|)$, $\mathcal{O}(m|D_j|)$, and $\mathcal{O}(m)$, where $m < |D_j|$. Not that, the computational cost of random features model is far less than $m^2|D_j|$. It is ignored when expressing the computational complexity. In the experiments, the training time of our methods includes the time of calculating the random features model.

The way of weighted averaging in Eq.(4) cannot improve the approximation ability of DRank-RF in each local processor (Huang & Huo, 2019; Lin et al., 2020b; Yin et al., 2021). To further improve the performance, we bring an efficient communication strategy into DRank-RF.

---

**Algorithm 1** Distributed Least Square Ranking with Random Features and communications (DRank-RF-C)

---

**Initialize**: $\bar{\mathbf{g}}^0_{m,D,\lambda} = \mathbf{0}$
**For** $l = 1$ **to** $M$ **do**
**Local processor**: compute the local gradient $G_{m,D_j,\lambda}(\bar{\mathbf{g}}^{l-1}_{m,D,\lambda})$ and communicate back to the global processor.
**Global processor**: compute $\bar{G}_{m,D,\lambda}(\bar{\mathbf{g}}^{l-1}_{m,D,\lambda}) = \sum_{j=1}^p \frac{|D_j|^2}{\sum_{k=1}^p |D_k|^2} G_{m,D_j,\lambda}(\bar{\mathbf{g}}^{l-1}_{m,D,\lambda})$ in Eq.(9) and communicate to each local processor.
**Local processor**: compute $\beta^{l-1}_j$ in Eq.(8) and communicate back to the global processor.
**Global processor**: compute $\bar{\mathbf{g}}^l_{m,D,\lambda}$ in Eq.(7), and communicate to each local processor.
**End For**
**Output**: $\bar{\mathbf{g}}^M_{m,D,\lambda}$ and $\bar{f}^M_{m,D,\lambda} = \langle \bar{\mathbf{g}}^M_{m,D,\lambda}, \phi_m(\cdot) \rangle$

---

## 3.2 DRANK-RF WITH COMMUNICATIONS (DRANK-RF-C)

In this section, we introduce the DRank-RF with communications (DRank-RF-C), which can not only improve the approximation ability but also protect the data privacy in each local processor.

For any $\mathbf{g}$, according to Eq.(3), one has the following equation:

$$
\begin{aligned}
\mathbf{g}_{m,D,\lambda} &= \mathbf{g} - (\mathbf{\Phi}_{m,D}\mathbf{W}_D\mathbf{\Phi}^T_{m,D} + \frac{\lambda}{2}\mathbf{I})^{-1}[(\mathbf{\Phi}_{m,D}\mathbf{W}_D\mathbf{\Phi}^T_{m,D} + \frac{\lambda}{2}\mathbf{I})\mathbf{g} - \mathbf{\Phi}_{m,D}\mathbf{W}_D\bar{\mathbf{y}}_D] \\
&= \mathbf{g} - (\mathbf{\Phi}_{m,D}\mathbf{W}_D\mathbf{\Phi}^T_{m,D} + \frac{\lambda}{2}\mathbf{I})^{-1}G_{m,D,\lambda}(\mathbf{g}),
\end{aligned}
\tag{5}
$$

where $G_{m,D,\lambda}(\mathbf{g}) = (\mathbf{\Phi}_{m,D}\mathbf{W}_D\mathbf{\Phi}^T_{m,D} + \frac{\lambda}{2}\mathbf{I})\mathbf{g} - \mathbf{\Phi}_{m,D}\mathbf{W}_D\bar{\mathbf{y}}_D$.

Define $\bar{\mathbf{g}}^0_{m,D,\lambda} = \sum_{j=1}^p \frac{|D_j|^2}{\sum_{k=1}^p |D_k|^2}\mathbf{g}_{m,D_j,\lambda}$, we can obtain that

$$
\bar{\mathbf{g}}^0_{m,D,\lambda} = \mathbf{g} - \sum_{j=1}^p \frac{|D_j|^2}{\sum_{k=1}^p |D_k|^2}(\mathbf{\Phi}_{m,D_j}\mathbf{W}_{D_j}\mathbf{\Phi}^T_{m,D_j} + \frac{\lambda}{2}\mathbf{I})^{-1}G_{m,D_j,\lambda}(\mathbf{g}).
\tag{6}
$$

Note that, the gradient of the empirical risk of $\frac{1}{|D_j|^2}\sum (y_i - y_k - (\mathbf{g}^T\phi_m(\mathbf{x}_i) - \mathbf{g}^T\phi_m(\mathbf{x}_k)))^2 + \lambda\|\mathbf{g}\|^2$ on $\mathbf{g}$ is $4G_{m,D_j,\lambda}(\mathbf{g})$ for all $(\mathbf{x}_i, y_i), (\mathbf{x}_k, y_k) \in D_j$.

Comparing Eq.(5) and Eq.(6), we consider the communication strategy based on the well-known Newton Raphson iteration (Lin et al., 2020b; Yin et al., 2021; Chen et al., 2021) for DRank-RF, which is formed as:

$$
\bar{\mathbf{g}}^l_{m,D,\lambda} = \bar{\mathbf{g}}^{l-1}_{m,D,\lambda} - \sum_{j=1}^p \frac{|D_j|^2}{\sum_{k=1}^p |D_k|^2}\beta^{l-1}_j,
\tag{7}
$$

where

$$
\beta^{l-1}_j = H^{-1}_{D_j,\lambda}\bar{G}_{m,D,\lambda}(\bar{\mathbf{g}}^{l-1}_{m,D,\lambda}),
\tag{8}
$$

$$
\bar{G}_{m,D,\lambda}(\mathbf{g}) = \sum_{j=1}^p \frac{|D_j|^2}{\sum_{k=1}^p |D_k|^2}G_{m,D_j,\lambda}(\mathbf{g}),
\tag{9}
$$

$H_{D_j,\lambda} = \mathbf{\Phi}_{m,D_j}\mathbf{W}_{D_j}\mathbf{\Phi}^T_{m,D_j} + \frac{\lambda}{2}\mathbf{I}$, and $l$ is the number of iteration.

The method of synthesis operation in DRank-RF-C is to weighted average the model parameters $\{\beta_j\}$ of each local processor obtained in the last iteration.

Algorithm 1 shows the detail of DRank-RF-C. In step 1, let $\bar{\mathbf{g}}^0_{m,D,\lambda}$ be $\mathbf{0}$. In the following steps, we update the global gradients and model parameters iteratively. For $l = 1, \ldots, M$, we distribute $\bar{\mathbf{g}}^{l-1}_{m,D,\lambda}$ to each local processor. In step 2 (on each local processor), compute $p$ local gradient vectors $G_{m,D_j,\lambda}(\bar{\mathbf{g}}^{l-1}_{m,D,\lambda})$ and communicate them back to the global processor. In step 3 (on global

processor), according to the received $p$ local gradient vectors, we compute the global gradient $\bar{G}_{m,D,\lambda}(\bar{\mathbf{g}}_{m,D,\lambda}^{l-1})$ and communicate it to each local processor. In step 4 (on each local processor), each local processor computes $\beta_j^{l-1}$ and communicates them back to the global processor. In step 5 (on global processor), the global processor obtains the solution $\bar{\mathbf{g}}_{m,D,\lambda}^l$. Then we transmit $\bar{\mathbf{g}}_{m,D,\lambda}^l$ to each local processor and go back to step 2.

**Complexity Analysis** In the terms of time complexity, one needs to compute the inverse of $\boldsymbol{\Phi}_{m,D_j}\mathbf{W}_{D_j}\boldsymbol{\Phi}_{m,D_j}^T + \frac{\lambda}{2}\mathbf{I}$ and the matrices multiplication $\boldsymbol{\Phi}_{m,D_j}\mathbf{W}_{D_j}\boldsymbol{\Phi}_{m,D_j}^T$ once for each local processor, and one needs to solve the local gradient $G_{m,D_j,\lambda}$ and model parameter $\beta_j$ in each iteration for each local processor. Thus, the total time complexity for each local processor is $\mathcal{O}(m^2|D_j| + mM|D_j|)$, where $M$ is the number of communications. In the terms of space complexity, for each local processor, the key is to store $\boldsymbol{\Phi}_{m,D_j}$, thus the space complexity of each local processor is $\mathcal{O}(m|D_j|)$. In the terms of communication complexity, the global processor sends the gradient $\bar{G}_{m,D,\lambda}$ and $\bar{\mathbf{g}}_{m,D,\lambda}^l$ to each local processor and receives the local gradient $G_{m,D_j,\lambda}$ and model parameter $\beta_j$ from each local processor in each iteration. Therefore, the total communication complexity is $\mathcal{O}(mM)$. Note that, if the number of communications $M \leq m$, the time complexity and space complexity of DRank-RF-C are the same as those of DRank-RF.

## 4 THEORETICAL ANALYSIS

Here, we analyze the convergence rate of DRank-RF and DRank-RF-C in probability. Define the optimal hypothesis $f_\lambda$ in $\mathcal{H}_K$ as $f_\lambda = \arg\min_{f \in \mathcal{H}_K} \left\{ \mathcal{E}(f) + \lambda\|f\|_K^2 \right\}$. We assume $f_\lambda$ exists.

### 4.1 CONVERGENCE RATE OF DRANK-RF

In the following, we state and discuss the convergence rate of DRank-RF in probability.

**Theorem 1.** *Suppose $\psi$ is continuous, such that $|\psi(\mathbf{x}, \boldsymbol{\omega})| \leq \tau$ almost surely, $\tau \in [1, \infty)$. Assume that $L_K^{-r}f_\rho \in \mathcal{H}_K$ with $0 < r \leq 1$, where $L_K^r$ is the $r$-th power of $L_K$. If the regularization parameter $\lambda = \mathcal{O}\left(\left(\sum_{j=1}^p \frac{|D_j|}{\sum_{k=1}^p |D_k|^2}\right)^{\frac{1}{1+r}}\right)$ and the number of random features $m = \Omega\left(\left(\sum_{j=1}^p \frac{|D_j|}{\sum_{k=1}^p |D_k|^2}\right)^{\frac{-2r}{1+r}}\right)$, for $\bar{f}_{m,D,\lambda}^0$ defined in Eq.(4) and every $\delta \in (0, 1]$, there holds $\left\|\bar{f}_{m,D,\lambda}^0 - f_\rho\right\|_K = \mathcal{O}\left(\left(\sum_{j=1}^p \frac{|D_j|}{\sum_{k=1}^p |D_k|^2}\right)^{\frac{r}{1+r}} \log \frac{1}{\delta}\right)$ with confidence at least $1 - \delta$.*

**Remark 1.** *From Theorem 1 mentioned above, one can see that if the number of random features $m$ is $\Omega\left(\left(\sum_{j=1}^p \frac{|D_j|}{\sum_{k=1}^p |D_k|^2}\right)^{\frac{-2r}{1+r}}\right)$, the convergence rate of the proposed DRank-RF can reach $\mathcal{O}\left(\left(\sum_{j=1}^p \frac{|D_j|}{\sum_{k=1}^p |D_k|^2}\right)^{\frac{r}{1+r}}\right)$[1], which is sharper than the existing convergence rate $\mathcal{O}\left(\left(\sum_{j=1}^p \frac{|D_j|^{3/2}}{\sum_{k=1}^p |D_k|^2}\right)^{\frac{r}{1+r}}\right)$ of the state-of-the-art distributed pairwise ranking kernel learning (Chen et al., 2021). When the number of partitions $p = 1$, the convergence rate of the proposed DRank-RF is $\mathcal{O}\left(|D|^{\frac{-r}{1+r}}\right)$ with $m = \Omega\left(|D|^{\frac{2r}{1+r}}\right)$. When $|D_1| = \ldots = |D_p|$, the convergence rate of the proposed DRank-RF is $\mathcal{O}\left((\frac{|D|}{p})^{\frac{-r}{1+r}}\right)$ with $m = \Omega\left((\frac{|D|}{p})^{\frac{2r}{1+r}}\right)$. Theoretical analysis demonstrates that the proposed DRank-RF is sound and effective.*

**Remark 2.** *From a theoretical perspective, this paper is a non-trivial extension of these approximate pairwise ranking methods. The existing papers mainly use capacity concentration estimation (Rudin, 2009; Rudin & Schapire, 2009; Rejchel, 2012) and algorithmic stability (Cossock & Zhang, 2008; Chen et al., 2014) for the learning theory analysis of pairwise ranking. In this paper, we apply the integral operator framework and introduce a novel technique of error decomposition so that the proposed method can achieve a tight bound under the basic condition. The details can be seen in*

---

[1]Logarithmic terms of convergence rates and complexity are hidden in this paper.

Table 1: The computational complexity of different algorithms. $m$ is the number of random features and $m < |D_j|$. $M$ is the number of communications. $q$ is the dimension of data. $|D_j| < |D|$.

| Algorithms | Time | Space | Communication |
|---|---|---|---|
| LSRank (Chen et al., 2019) | $|D|^3$ | $|D|^2$ | / |
| DRank(Chen et al., 2021; 2019) | $|D_j|^3$ | $|D_j|^2$ | $|D_j|$ |
| DRank-C(Chen et al., 2021) | $|D_j|^3 + M|D_j||D|$ | $|D_j|^2$ | $qM|D|$ |
| **DRank-RF (This Paper)** | $m^2|D_j|$ | $m|D_j|$ | $m$ |
| **DRank-RF-C (This Paper)** | $m^2|D_j| + mM|D_j|$ | $m|D_j|$ | $mM$ |

*Appendix. This is the first time that combined distributed learning and random features in LSRank and achieved such a breakthrough.*

### 4.2 CONVERGENCE RATE OF DRANK-RF-C

Here, we introduce and discuss the convergence analysis of DRank-RF-C in probability.

**Theorem 2.** *Suppose $\psi$ is continuous, such that $|\psi(\mathbf{x}, \boldsymbol{\omega})| \leq \tau$ almost surely, $\tau \in [1, \infty)$. Assume that $L_K^{-r} f_\rho \in \mathcal{H}_K$ with $0 < r \leq 1$, where $L_K^r$ is the $r$-th power of $L_K$. If $\lambda = \mathcal{O}(|D|^{-\frac{1}{1+r}})$, $|D_1| = \ldots = |D_p| = \frac{|D|}{p}$, and the number of random features $m = \Omega\left(|D|^{\frac{2r}{1+r}}\right)$, for every $\delta \in (0, 1]$, with confidence at least $1 - \delta$, we have $\left\| \bar{f}_{m,D,\lambda}^M - f_\rho \right\|_K = \mathcal{O}\left( \left( p^{\frac{1}{2}} |D|^{-\frac{r}{2(1+r)}} \right)^{M+2} \right)$.*

*Proof.* The proof of Theorem 1 and 2 is in Appendix. $\qquad\square$

The assumption of $L_K^{-r} f_\rho \in \mathcal{H}_K$ with $0 < r \leq 1$ is commonly used in approximation theory (Smale & Zhou, 2007), which can be seen as regularity assumption.

**Remark 3.** *Theoretical analysis shows that, when $p < |D|^{\frac{rM}{rM+M+2}}$, the convergence rate of DRank-RF-C is sharper than that of DRank-RF at the same settings. Note that $p$ is monotonically increasing with the number of communications $M$, which can demonstrate the power of the proposed communications. For $M \to \infty$, it is clear that the convergence rate of DRank-RF-C is always sharper than that of DRank-RF. The convergence rate in Theorem 2 is also related to $\delta$. To simplify the representation, we omit it here. Their detailed relationship is shown in Appendix C.2.*

## 5 COMPARED WITH THE RELATED WORKS

In this section, we introduce the related distributed pairwise ranking in kernel learning. In Chen et al. (2019), Chen et al. construct the divide-and-conquer pairwise ranking in kernel learning, called DRank. They study the statistical properties of DRank and establish its convergence analysis in expectation. The time complexity, space complexity, and communication complexity of DRank are $\mathcal{O}(|D_j|^3)$, $\mathcal{O}(|D_j|^2)$, and $\mathcal{O}(|D_j|)$, respectively. The convergence rate in expectation only demonstrates the average information for multiple trails but fails to capture the learning performance for a single trail. Therefore, the probability version of the convergence rate of DRank in a single trial is proposed subsequently in Chen et al. (2021). The statistical properties and the convergence rate of DRank in probability are carefully analyzed and established in Chen et al. (2021). In addition, the paper Chen et al. (2021) proposes a communication strategy for DRank, called DRank-C, to improve the learning performance and provides its convergence rate in probability. The time complexity and space complexity of DRank-C are $\mathcal{O}(|D_j|^3 + M|D_j||D|)$ and $\mathcal{O}(|D_j|^2)$, respectively. However, its communication strategy requires communicating the input data between each local processor. Thus, it is difficult to protect the data privacy of each local processor. Furthermore, for each iteration, the communication complexity of each local processor is $\mathcal{O}(qM|D|)$, where $q$ denotes the dimension, which is infeasible in practice for large-scale datasets.

Table 1 shows the detail complexity of the related methods. We see that the proposed DRank-RF only requires $\mathcal{O}(m^2|D_j|)$ time, $\mathcal{O}(m|D_j|)$ memory, and $\mathcal{O}(m)$ communications, which are smaller

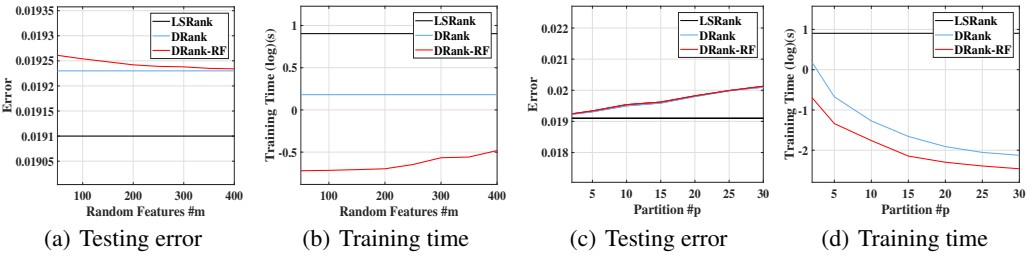

Figure 1: The testing error and training time on simulated datasets. (a) and (b) are about the number of random features $m$ with $p = 2$. (c) and (d) are about the number of partitions $p$ with $m = 200$ in DRank-RF.

than other methods. For DRank-RF-C, it requires less complexity than the communication-based method. In addition, the communication strategy proposed in this paper only requires communicating the gradient and the model parameters, rather than the data, therefore the proposed DRank-RF-C do better on privacy protection.

The convergence rate of the proposed DRank-RF in Theorem 1 is sharper than the convergence rate $\mathcal{O}\left(\left(\sum_{j=1}^{p}\frac{|D_j|^{3/2}}{\sum_{k=1}^{p}|D_k|^2}\right)^{\frac{r}{1+r}}\right)$ of the existing state-of-the-art DRank (without communications) (Chen et al., 2021; 2019). And the convergence rate of the proposed DRank-RF-C in Theorem 2 is also sharper than the convergence rate $\mathcal{O}\left(\max\left\{\left(p^{\frac{1}{2}}|D|^{-\frac{r}{2(1+r)}}\right)^{M+1}, |D|^{-\frac{r}{2(1+r)}}\right\}\right)$ of the existing communication-based DRank (Chen et al., 2021).

## 6 EMPIRICAL EVALUATIONS

We perform experiments to validate our theoretical analysis of DRank-RF and the communication strategy on simulated and real datasets. The server is 32 cores (2.40GHz) and 32 GB of RAM.

### 6.1 PARAMETERS AND CRITERION

We use the Gaussian kernel $K(\mathbf{x}, \mathbf{x}') = \exp\left(-\|\mathbf{x} - \mathbf{x}'\|_2^2 / (2d^2)\right)$. The optimal bandwidth $d \in 2^{[-2:0.5:5]}$ and regularization parameter $\lambda \in 2^{[-13:2:-3]}$ are selected via 5-fold cross-validation. The criterion of evaluating the methods on testing data is as follows (Chen et al., 2021; Kriukova et al., 2016): $\mathcal{R}(f) = \frac{\sum_{i,j=1}^{n'} I_{\{(y_i > y_j) \wedge (\bar{f}(\mathbf{x}_i) \leq \bar{f}(\mathbf{x}_j))\}}}{\sum_{i,j=1}^{n'} I_{\{y_i > y_j\}}}$, where $I_{\{\varphi\}}$ is 1 if $\varphi$ is true and 0 otherwise.

We use the exact LSRank as a baseline, which trains all samples in a batch. And we compare the proposed DRank-RF and DRank-RF-C ($M = 2, 4, 8$) with DRank, DRank-C, and LSRank by carrying out various settings. We repeat the training 5 times and estimate the error on testing data.

### 6.2 SIMULATED EXPERIMENTS

Inspired by the numerical experiments in Chen et al. (2021); Kriukova et al. (2016), we consider the following way to generate the synthetic data. The inputs $\{\mathbf{x}_i\}_{i=1}^{|D'|} \in \mathbb{R}^{|D'| \times q}$ are randomly chosen from $\{1, \cdots, 100\}$, and the corresponding outputs $\{y_i\}_{i=1}^{|D'|}$ are generated from the model $y_i = [\|\mathbf{x}_i\| / 7] + \epsilon_i, \quad 1 \leq i \leq |D'|$, where $[\cdot]$ means the integer part of inputs and $\epsilon_i$ is the noise independently sampled from Gaussian distribution $\mathcal{N}(0, 0.01)$. Dimension $q$ is 7. We generate 20000 samples. 70% is used for training and 30% for testing.

Figure 1(a) and Figure 1(b) show the testing error and training time (logarithmizing it) in seconds about the number of random features $m$ with $p = 2$ and indicate that DRank-RF has an obvious advantage over DRank and LSRank, even one order of magnitude less, in time cost. In the testing error, the gap between DRank-RF and DRank decreases as $m$ increases. Finally, there is no significant dif-

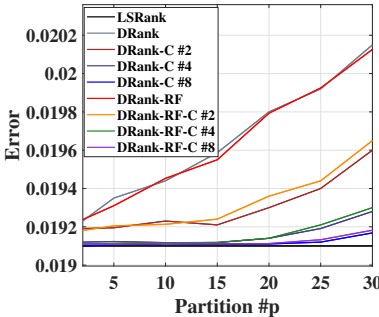

Figure 2: The testing error about the number of partitions $p$ on simulated datasets. 2, 4, and 8 represent the number of communications with $m = 300$.

ference between DRank-RF and DRank, both of which are close to the optimal level. These results are consistent with our theoretical analysis. With the increase of the number of random features $m$, the training time of DRank-RF increases, and the testing error becomes smaller, which are in line with the theoretical reasoning. And the testing error of DRank-RF declines significantly when $m$ is a small number. Therefore, in practice, we only need to take a small $m$ to obtain a satisfactory error, which will result in the savings of computing resources. Note that, DRank and LSRank have nothing to do with $m$.

Figure 1(c) and Figure 1(d) show the testing error and training time about the number of partitions $p$ with $m = 200$ for DRank-RF. Figure 1(c) shows DRank-RF keeps the same accuracy level as DRank. With the increase of the number of partitions $p$, the testing errors increase in $p$-related algorithms, which is in line with the theoretical analysis. In Figure 1(d), with the increase of $p$, the training time decreases in distributed algorithms (DRank-RF and DRank). Our algorithm DRank-RF has a significant advantage over LSRank and DRank in the training time. In particular, the time cost of DRank with $p = 30$ is higher than that of DRank-RF with $p = 15$, that is to say, the proposed DRank-RF requires less expensive hardware devices, under the same scenario and time cost. Combining Figure 1(c) and Figure 1(d), we obtain that DRank-RF can use fewer hardware devices (local processors) to achieve a smaller testing error under the same training time, which is consistent with the theoretical analysis.

Figure 2 shows the relation between the testing error, $p$, and different numbers of communications ($M = 2, 4, 8$) with $m = 300$ and indicates the following information: 1) With the increase of $p$, the testing error gaps between $p$-related algorithms and exact LSRank become larger and larger. There exists an upper bound of $p$ for DRank-RF and DRank-RF-C respectively, when larger than it, the testing error increases and is far from the exact LSRank. This is in line with Theorem 1 and Theorem 2. 2) The upper bound $p$ of DRank-RF-C is much larger than DRank-RF, which is aligned with our theoretical analysis that the bound of $p$ is determined by the communication times. 3) Under the same $p$, the performance of DRank-RF-C is better than DRank-RF. And with the increase of the number of communications $M$, the testing error of DRank-RF-C is smaller. These verify the power of the communication strategy for DRank-RF. 4) Under the same conditions, the testing errors of the proposed DRank-RF and DRank-RF-C are similar to those of DRank and DRank-C.

## 6.3  REAL DATA

The real dataset of MovieLens is from website http://www.grouplens.org/taxonomy/term/14, which is a $62423 \times 162541$ rating matrix where $(i, j)$ entry is the rating score of the $j$-th reviewer on the $i$-th movie. We group the reviewers into 500-1000 movies according to the number of movies they have rated. And 500 reference reviewers are selected at random from this group. In addition, we select the test reviewers from those users who had rated more than 5000 movies. So, we obtain a small matrix with the scale of at least $5000 \times 501$, where the last column corresponds to the test reviewer and the other columns correspond to the 500 reference reviewers. Then the columns without non-zero elements are deleted and the rows without the rating of any reference reviewers or the test reviewer are deleted. Finally, missing review values of every left movie were replaced

Table 2: Comparison of the average testing error (standard deviation) and training time (in seconds) on Movie-Lens dataset, with partitions $p = 2, 10, 15$ and random features $m = 100, 150$. 2, 8, and 16 are the number of communications.

| Algorithm (m=100) | p=2 | | p=10 | | p=15 | |
|---|---|---|---|---|---|---|
| | Error | Time | Error | Time | Error | Time |
| LSRank | $0.4902 \pm 0.0283$ | 4.01 | $0.4902 \pm 0.0283$ | 4.01 | $0.4902 \pm 0.0283$ | 4.01 |
| DRank | $0.4904 \pm 0.0219$ | 2.35 | $0.4906 \pm 0.0220$ | 0.08 | $0.4907 \pm 0.0222$ | 0.05 |
| DRank-C #2 | $0.4904 \pm 0.0221$ | 2.73 | $0.4905 \pm 0.0219$ | 0.10 | $0.4906 \pm 0.0181$ | 0.08 |
| DRank-C #8 | $0.4903 \pm 0.0192$ | 3.69 | $0.4903 \pm 0.0192$ | 0.19 | $0.4905 \pm 0.0212$ | 0.10 |
| DRank-RF | $0.4904 \pm 0.0221$ | 0.16 | $0.4907 \pm 0.0211$ | 0.02 | $0.4908 \pm 0.0199$ | 0.01 |
| DRank-RF-C #2 | $0.4904 \pm 0.0210$ | 0.22 | $0.4906 \pm 0.0171$ | 0.03 | $0.4907 \pm 0.0217$ | 0.02 |
| DRank-RF-C #8 | $0.4903 \pm 0.0187$ | 0.32 | $0.4903 \pm 0.0210$ | 0.03 | $0.4905 \pm 0.0211$ | 0.02 |
| DRank-RF-C #16 | $0.4903 \pm 0.0103$ | 0.41 | $0.4903 \pm 0.0185$ | 0.04 | $0.4904 \pm 0.0236$ | 0.03 |
| Algorithm (m=150) | p=2 | | p=10 | | p=15 | |
| | Error | Time | Error | Time | Error | Time |
| DRank-RF | $0.4904 \pm 0.0201$ | 0.17 | $0.4906 \pm 0.0197$ | 0.03 | $0.4907 \pm 0.0232$ | 0.01 |
| DRank-RF-C #2 | $0.4904 \pm 0.0191$ | 0.23 | $0.4905 \pm 0.0197$ | 0.04 | $0.4906 \pm 0.0180$ | 0.01 |
| DRank-RF-C #8 | $0.4903 \pm 0.0167$ | 0.35 | $0.4903 \pm 0.0187$ | 0.05 | $0.4904 \pm 0.0221$ | 0.02 |
| DRank-RF-C #16 | $0.4903 \pm 0.0092$ | 0.44 | $0.4903 \pm 0.0121$ | 0.06 | $0.4904 \pm 0.0111$ | 0.03 |

with the median review score of those left reference reviewers on this movie. Here, we obtain a smaller matrix. Each row of it is a data pair $(\mathbf{x}_i, y_i)$ and the last entry was the label $y_i$ of the input features $\mathbf{x}_i$. The experimental dataset is similar to that in Chen et al. (2021). On the obtained dataset, 70% is used for training and 30% for testing. The empirical evaluations are given in Table 2 where $m = 100, 150$ and $p = 2, 10, 15$. In Table 2, we can find that the experimental results are similar to those on the simulated data. The average testing error gaps between our methods and the exact methods are particularly small, which verify the effectiveness of our methods on the real dataset. Under the conditions of $M$=16, $p$=2, and $p$=10, the testing error of DRank-RF-C is convergent and does not change with the increase of the number of communications. Under the condition of $p$=15, the testing error of DRank-RF-C decreases with the increase of the number of communications, which demonstrates the effectiveness of the communication strategy on the real dataset and is consistent with our Theorem 2. The training time in the distributed algorithms decreases with the increase of $p$. The training time in communication-based algorithms increases with the increase of the number of communications. The proposed DRank-RF and Drank-RF-C have significant advantages over LSRank, DRank, and DRank-C in the training time. These are consistent with the theoretical analysis. More experiments on different datasets are given in Appendix E.

# 7 CONCLUSIONS

We propose a novel pairwise ranking method (DRank-RF) to scale to large-scale scenarios. Our work is the first time to apply random features to least square ranking, which is a new exploration of the application of random features. Our theoretical analysis based on the techniques of integral operators shows that its convergence rate is sharper than that of the existing state-of-the-art DRank without communications. In computational complexity, DRank-RF only requires $\mathcal{O}(m^2|D_j|)$ time and $\mathcal{O}(m|D_j|)$ memory, which are the least compared with the existing state-of-the-art DRank. Experiments verify that our proposed method keeps the similar testing error as the exact and state-of-the-art approximate methods and has a greatly advantage over the exact and state-of-the-art approximate methods in the training time, which are consistent with our theoretical analysis. To further improve the performance of DRank-RF, we propose a communication strategy to DRank-RF, which is called DRank-RF-C. Statistical analysis shows that DRank-RF-C obtains a faster convergence rate than DRank-RF. Compared with the existing state-of-the-art DRank with communications, DRank-RF-C requires less complexity and keeps a sharper convergence rate. And the numerical results validate the power of the communication strategy.

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

## A    PRELIMINARY DEFINITIONS

There is a compact metric space $\mathcal{Z} := (\mathcal{X}, \mathcal{Y}) \subset \mathbb{R}^{q+1}$, where $\mathcal{X} \subset \mathbb{R}^q$ and $\mathcal{Y} \subset [-b, b]$ for some positive constant $b$. The sample set $D := \{(\mathbf{x}_i, y_i)\}_{i=1}^{N}$ of size $N = |D|$ is drawn independently from an intrinsic Borel probability measure $\rho$ on $\mathcal{Z}$. $\rho(y|X = \mathbf{x})$ denotes the conditional distribution for given input $\mathbf{x}$. The hypothesis space used is the reproducing kernel Hilbert space (RKHS) $(\mathcal{H}_K)$ associated with a mercer kernel $K : \mathcal{X} \times \mathcal{X} \to \mathbb{R}$ (Aronszajn, 1950; Cucker & Zhou, 2007). We will denote the inner product in $\mathcal{H}_K$ by $\langle \cdot, \cdot \rangle$, and corresponding norm by $\| \cdot \|_K$. $K_{\mathbf{x}} = K(\mathbf{x}, \cdot)$. Let $\rho_{\mathcal{X}}$ be the margin distribution of $\rho$ with respect to $\mathcal{X}$ and $L^2_{\rho_{\mathcal{X}}}$ be the Hilbert space of $\rho_{\mathcal{X}}$ square integrable functions on $\mathcal{X}$.

The Mercer kernel $K$ defines an integral operator $L_K$ on $\mathcal{H}_K$ (or $L^2_{\rho_{\mathcal{X}}}$) (Chen et al., 2021) by

$$L_K f = \int_{\mathcal{X}} \int_{\mathcal{X}} f(\mathbf{x}) \left( K_{\mathbf{x}} - K_{\mathbf{x}'} \right) d\rho_{\mathcal{X}}(\mathbf{x}) d\rho_{\mathcal{X}}(\mathbf{x}') .$$

Suppose $\psi$ is continuous, such that $|\psi(\mathbf{x}, \boldsymbol{\omega})| \le \tau$ almost surely, $\tau \in [1, \infty)$. Assume that $L_K^{-r} f_\rho \in \mathcal{H}_K$ with $0 < r \le 1$, where $L_K^r$ is the $r$-th power of $L_K$.

Before the proof, we give some definitions: $S_m : \mathbb{R}^m \to L^2_{\rho_{\mathcal{X}}}, (S_m \mathbf{g})(\mathbf{x}) = \langle \mathbf{g}, \phi_m(\mathbf{x}) \rangle$, $S_m^* : L^2_{\rho_{\mathcal{X}}} \to \mathbb{R}^m, S_m^* f = \int_{\mathcal{X}} \phi_m(\mathbf{x}) f(\mathbf{x}) d\rho_{\mathcal{X}}(\mathbf{x}), S_{m,D}^* : L^2_{\rho_{\mathcal{X}}} \to \mathbb{R}^m, S_{m,D}^* f = \frac{1}{|D|} \sum_{\mathbf{x}_j \in D_{\mathcal{X}}} \phi_m(\mathbf{x}_j) f(\mathbf{x}_j)$. $S_m^* S_m$ and $\boldsymbol{\Phi}_{m,D} \boldsymbol{\Phi}_{m,D}^{\mathrm{T}} = S_{m,D}^* S_m$ are self-adjoint and positive operators, with spectrum is $[0, \tau^2]$ (Caponnetto & Vito, 2007).

This part is organized as follows: In section B, we introduce the proof of Theorem 1. Section B.1 contains the main lemmas used for the proof of Theorem 1 and Theorem 2. Section B.2 is the detail proof process of Theorem 1. In section C, we introduce the proof of Theorem 2. Section C.1 contains the main lemmas used for the proof of Theorem 2. Section C.2 is the detail proof process of Theorem 2. In section D, we introduce the propositions used for the proof of Theorem 1 and Theorem 2. Section E is the experiments on Jester Joke dataset.

## B    PROOF OF THEOREM 1

### B.1    BOUND TERMS

**Lemma 1.** *We have*

$$\sqrt{\lambda} \|\mathbf{g}_{m,D,\lambda} - \mathbf{g}_{m,\lambda}\|$$

$$\le \sqrt{2} \left\| \left( \boldsymbol{\Phi}_{m,D} \mathbf{W}_D \boldsymbol{\Phi}_{m,D}^T + \frac{\lambda}{2} I \right)^{-1/2} \left( S_m^* S_m + \frac{\lambda}{2} I \right)^{1/2} \right\|$$

$$* \left( \left\| \left( S_m^* S_m + \frac{\lambda}{2} I \right)^{-1/2} (\boldsymbol{\Phi}_{m,D} \mathbf{W}_D \bar{\mathbf{y}}_D - S_m^* \mathbf{W}_D f_\rho) \right\| + \left\| \left( S_m^* S_m + \frac{\lambda}{2} I \right)^{-1/2} (S_m^* \mathbf{W}_D f_\rho - S_{m,D}^* \mathbf{W}_D f_\rho) \right\| \right)$$

$$+ \left( 1 + \left\| \left( \boldsymbol{\Phi}_{m,D} \mathbf{W}_D \boldsymbol{\Phi}_{m,D}^T + \frac{\lambda}{2} I \right)^{-1/2} \left( S_m^* S_m + \frac{\lambda}{2} I \right)^{1/2} \right\| \right) \|f_{m,\lambda} - f_\rho\|_K .$$

*Proof.* Note that $\|\mathbf{g}_{m,D,\lambda} - \mathbf{g}_{m,\lambda}\| \le \left\| \mathbf{g}_{m,D,\lambda} - \mathbf{g}_{m,D,\lambda}^{\diamond} \right\| + \left\| \mathbf{g}_{m,D,\lambda}^{\diamond} - \mathbf{g}_{m,\lambda} \right\|$. Define $f_{m,D,\lambda} = \mathbf{g}_{m,D,\lambda}^{\mathrm{T}} \phi_m(\cdot)$,

$$\mathbf{g}_{m,D,\lambda} = \underset{\mathbf{g} \in \mathbb{R}^m}{\arg\min} \left\{ \frac{1}{|D|^2} \sum_{z_i, z_k \in D} \left( (\mathbf{g}^{\mathrm{T}} \phi_m(\mathbf{x}_i) - y_i) - (\mathbf{g}^{\mathrm{T}} \phi_m(\mathbf{x}_k) - y_k) \right)^2 + \lambda \|\mathbf{g}\|^2 \right\},$$

$f_{m,D,\lambda}^{\diamond} = \mathbf{g}_{m,D,\lambda}^{\diamond \mathrm{T}} \phi_m(\cdot)$,

$$\mathbf{g}_{m,D,\lambda}^{\diamond} = \underset{\mathbf{g} \in \mathbb{R}^m}{\arg\min} \left\{ \frac{1}{|D|^2} \sum_{z_i, z_k \in D} \left( (\mathbf{g}^{\mathrm{T}} \phi_m(\mathbf{x}_i) - f_\rho(\mathbf{x}_i)) - (\mathbf{g}^{\mathrm{T}} \phi_m(\mathbf{x}_k) - f_\rho(\mathbf{x}_k)) \right)^2 + \lambda \|\mathbf{g}\|^2 \right\}.$$

One can have $f_{m,D,\lambda} = S_m \mathbf{g}_{m,D,\lambda}$, $\mathbf{g}_{m,D,\lambda} = \left( \boldsymbol{\Phi}_{m,D} \mathbf{W}_D \boldsymbol{\Phi}_{m,D}^T + \frac{\lambda}{2} I \right)^{-1} \boldsymbol{\Phi}_{m,D} \mathbf{W}_D \bar{\mathbf{y}}_D$,

$f^{\diamond}_{m,D,\lambda} = S_m \mathbf{g}^{\diamond}_{m,D,\lambda}$, and $\mathbf{g}^{\diamond}_{m,D,\lambda} = \left( \mathbf{\Phi}_{m,D} \mathbf{W}_D \mathbf{\Phi}^T_{m,D} + \frac{\lambda}{2} I \right)^{-1} S^*_{m,D} \mathbf{W}_D f_{\rho}$, so we have

$$
\begin{aligned}
& \mathbf{g}_{m,D,\lambda} - \mathbf{g}^{\diamond}_{m,D,\lambda} \\
&= \left( \mathbf{\Phi}_{m,D} \mathbf{W}_D \mathbf{\Phi}^T_{m,D} + \frac{\lambda}{2} I \right)^{-1} \left( \mathbf{\Phi}_{m,D} \mathbf{W}_D \bar{\mathbf{y}}_D - S^*_{m,D} \mathbf{W}_D f_{\rho} \right) \\
&= \left( \mathbf{\Phi}_{m,D} \mathbf{W}_D \mathbf{\Phi}^T_{m,D} + \frac{\lambda}{2} I \right)^{-1/2} \left( \mathbf{\Phi}_{m,D} \mathbf{W}_D \mathbf{\Phi}^T_{m,D} + \frac{\lambda}{2} I \right)^{-1/2} \left( S^*_m S_m + \frac{\lambda}{2} I \right)^{1/2} \quad (10) \\
& \quad * \left( S^*_m S_m + \frac{\lambda}{2} I \right)^{-1/2} \left( \mathbf{\Phi}_{m,D} \mathbf{W}_D \bar{\mathbf{y}}_D - S^*_{m,D} \mathbf{W}_D f_{\rho} \right).
\end{aligned}
$$

Note that $\left\| \left( \mathbf{\Phi}_{m,D} \mathbf{W}_D \mathbf{\Phi}^T_{m,D} + \frac{\lambda}{2} I \right)^{-1/2} \right\| \leq \sqrt{2/\lambda}$. Thus we can obtain that

$$
\begin{aligned}
& \left\| \mathbf{g}_{m,D,\lambda} - \mathbf{g}^{\diamond}_{m,D,\lambda} \right\| \\
&\leq \sqrt{2/\lambda} \left\| \left( \mathbf{\Phi}_{m,D} \mathbf{W}_D \mathbf{\Phi}^T_{m,D} + \frac{\lambda}{2} I \right)^{-1/2} \left( S^*_m S_m + \frac{\lambda}{2} I \right)^{1/2} \right\| \\
& \quad * \left\| \left( S^*_m S_m + \frac{\lambda}{2} I \right)^{-1/2} \left( \mathbf{\Phi}_{m,D} \mathbf{W}_D \bar{\mathbf{y}}_D - S^*_{m,D} \mathbf{W}_D f_{\rho} \right) \right\| \\
&= \sqrt{2/\lambda} \left\| \left( \mathbf{\Phi}_{m,D} \mathbf{W}_D \mathbf{\Phi}^T_{m,D} + \frac{\lambda}{2} I \right)^{-1/2} \left( S^*_m S_m + \frac{\lambda}{2} I \right)^{1/2} \right\| \\
& \quad * \left\| \left( S^*_m S_m + \frac{\lambda}{2} I \right)^{-1/2} \left( \mathbf{\Phi}_{m,D} \mathbf{W}_D \bar{\mathbf{y}}_D - S^*_m \mathbf{W}_D f_{\rho} + S^*_m \mathbf{W}_D f_{\rho} - S^*_{m,D} \mathbf{W}_D f_{\rho} \right) \right\| \\
&\leq \sqrt{2/\lambda} \left\| \left( \mathbf{\Phi}_{m,D} \mathbf{W}_D \mathbf{\Phi}^T_{m,D} + \frac{\lambda}{2} I \right)^{-1/2} \left( S^*_m S_m + \frac{\lambda}{2} I \right)^{1/2} \right\| \\
& \quad * \left( \left\| \left( S^*_m S_m + \frac{\lambda}{2} I \right)^{-1/2} \left( \mathbf{\Phi}_{m,D} \mathbf{W}_D \bar{\mathbf{y}}_D - S^*_m \mathbf{W}_D f_{\rho} \right) \right\| + \left\| \left( S^*_m S_m + \frac{\lambda}{2} I \right)^{-1/2} \left( S^*_m \mathbf{W}_D f_{\rho} - S^*_{m,D} \mathbf{W}_D f_{\rho} \right) \right\| \right).
\end{aligned}
$$
$$(11)$$

Define $f_{m,\lambda} = \mathbf{g}^T_{m,\lambda} \boldsymbol{\phi}_m(\cdot)$ with

$$
\mathbf{g}_{m,\lambda} = \underset{\mathbf{g} \in \mathbb{R}^m}{\arg\min} \left\{ \int_{\mathcal{Z}} \int_{\mathcal{Z}} \left( (\mathbf{g}^T \boldsymbol{\phi}_m(\mathbf{x}) - f_{\rho}(\mathbf{x})) - (\mathbf{g}^T \boldsymbol{\phi}_m(\mathbf{x}') - f_{\rho}(\mathbf{x}')) \right)^2 d\rho_{\mathcal{X}}(\mathbf{x}, y) d\rho_{\mathcal{X}}(\mathbf{x}', y') + \lambda \|\mathbf{g}\|^2 \right\}.
$$

We know $f_{m,\lambda} = S_m \mathbf{g}_{m,\lambda}$ and $\mathbf{g}_{m,\lambda} = \left( S^*_m \mathbf{W}_D S_m + \frac{\lambda}{2} I \right)^{-1} S^*_m \mathbf{W}_D f_{\rho}$. So one can obtain

$$
\begin{aligned}
& \mathbf{g}^{\diamond}_{m,D,\lambda} - \mathbf{g}_{m,\lambda} \\
&= \left( \mathbf{\Phi}_{m,D} \mathbf{W}_D \mathbf{\Phi}^T_{m,D} + \frac{\lambda}{2} I \right)^{-1} S^*_{m,D} \mathbf{W}_D f_{\rho} - \left( S^*_m \mathbf{W}_D S_m + \frac{\lambda}{2} I \right)^{-1} S^*_m \mathbf{W}_D f_{\rho} \\
&= \left( \mathbf{\Phi}_{m,D} \mathbf{W}_D \mathbf{\Phi}^T_{m,D} + \frac{\lambda}{2} I \right)^{-1} \left[ S^*_{m,D} \mathbf{W}_D f_{\rho} - S^*_m \mathbf{W}_D f_{\rho} \right] \\
& \quad + \left[ \left( \mathbf{\Phi}_{m,D} \mathbf{W}_D \mathbf{\Phi}^T_{m,D} + \frac{\lambda}{2} I \right)^{-1} - \left( S^*_m \mathbf{W}_D S_m + \frac{\lambda}{2} I \right)^{-1} \right] S^*_m \mathbf{W}_D f_{\rho}.
\end{aligned}
$$

For any self-adjoint and positive operators $A$ and $B$,

$$
A^{-1} - B^{-1} = A^{-1}(B - A)B^{-1}, A^{-1} - B^{-1} = B^{-1}(B - A)A^{-1},
$$

so we have

$$\mathbf{g}_{m,D,\lambda}^{\diamond} - \mathbf{g}_{m,\lambda} = \left( \boldsymbol{\Phi}_{m,D} \mathbf{W}_D \boldsymbol{\Phi}_{m,D}^T + \frac{\lambda}{2}I \right)^{-1} \left[ S_{m,D}^* \mathbf{W}_D f_\rho - S_m^* \mathbf{W}_D f_\rho \right]$$

$$+ \left( \boldsymbol{\Phi}_{m,D} \mathbf{W}_D \boldsymbol{\Phi}_{m,D}^T + \frac{\lambda}{2}I \right)^{-1} \left( S_m^* \mathbf{W}_D S_m - \boldsymbol{\Phi}_{m,D} \mathbf{W}_D \boldsymbol{\Phi}_{m,D}^T \right) \mathbf{g}_{m,\lambda}$$

$$< \left( \boldsymbol{\Phi}_{m,D} \mathbf{W}_D \boldsymbol{\Phi}_{m,D}^T + \frac{\lambda}{2}I \right)^{-1} \left[ S_{m,D}^* \mathbf{W}_D f_\rho - S_m^* \mathbf{W}_D f_\rho \right]$$

$$+ \left( \boldsymbol{\Phi}_{m,D} \mathbf{W}_D \boldsymbol{\Phi}_{m,D}^T + \frac{\lambda}{2}I \right)^{-1} \left( S_m^* S_m - \boldsymbol{\Phi}_{m,D} \boldsymbol{\Phi}_{m,D}^T \right) \mathbf{g}_{m,\lambda}.$$

We know that $\boldsymbol{\Phi}_{m,D} \boldsymbol{\Phi}_{m,D}^{\mathrm{T}} = S_{m,D}^* S_m$ (Caponnetto & Vito, 2007), thus we can obtain that

$$\mathbf{g}_{m,D,\lambda}^{\diamond} - \mathbf{g}_{m,\lambda}$$

$$< \left( \boldsymbol{\Phi}_{m,D} \mathbf{W}_D \boldsymbol{\Phi}_{m,D}^T + \frac{\lambda}{2}I \right)^{-1} \left[ S_{m,D}^* \mathbf{W}_D f_\rho - S_m^* \mathbf{W}_D f_\rho \right]$$

$$+ \left( \boldsymbol{\Phi}_{m,D} \mathbf{W}_D \boldsymbol{\Phi}_{m,D}^T + \frac{\lambda}{2}I \right)^{-1} \left( S_m^* S_m \mathbf{g}_{m,\lambda} - S_{m,D}^* S_m \mathbf{g}_{m,\lambda} \right)$$

$$\leq \left( \boldsymbol{\Phi}_{m,D} \mathbf{W}_D \boldsymbol{\Phi}_{m,D}^T + \frac{\lambda}{2}I \right)^{-1} \left[ S_{m,D}^* f_\rho - S_{m,D}^* S_m \mathbf{g}_{m,\lambda} \right]$$

$$+ \left( \boldsymbol{\Phi}_{m,D} \mathbf{W}_D \boldsymbol{\Phi}_{m,D}^T + \frac{\lambda}{2}I \right)^{-1} \left[ S_m^* S_m \mathbf{g}_{m,\lambda} - S_m^* f_\rho \right]$$

$$= \left( \boldsymbol{\Phi}_{m,D} \mathbf{W}_D \boldsymbol{\Phi}_{m,D}^T + \frac{\lambda}{2}I \right)^{-1} \left[ S_{m,D}^* f_\rho - S_{m,D}^* f_{m,\lambda} \right] + \left( \boldsymbol{\Phi}_{m,D} \mathbf{W}_D \boldsymbol{\Phi}_{m,D}^T + \frac{\lambda}{2}I \right)^{-1} \left[ S_m^* f_{m,\lambda} - S_m^* f_\rho \right]$$

$$= \left( \boldsymbol{\Phi}_{m,D} \mathbf{W}_D \boldsymbol{\Phi}_{m,D}^T + \frac{\lambda}{2}I \right)^{-1} S_{m,D}^* \left[ f_\rho - f_{m,\lambda} \right] + \left( \boldsymbol{\Phi}_{m,D} \mathbf{W}_D \boldsymbol{\Phi}_{m,D}^T + \frac{\lambda}{2}I \right)^{-1} S_m^* \left[ f_{m,\lambda} - f_\rho \right].$$

$$\tag{12}$$

Thus, we have

$$\left\| \mathbf{g}_{m,D,\lambda}^{\diamond} - \mathbf{g}_{m,\lambda} \right\|$$

$$\leq \left( \left\| \left( \boldsymbol{\Phi}_{m,D} \mathbf{W}_D \boldsymbol{\Phi}_{m,D}^T + \frac{\lambda}{2}I \right)^{-1} S_{m,D}^* \right\| + \left\| \left( \boldsymbol{\Phi}_{m,D} \mathbf{W}_D \boldsymbol{\Phi}_{m,D}^T + \frac{\lambda}{2}I \right)^{-1} S_m^* \right\| \right) \left\| f_{m,\lambda} - f_\rho \right\|_K.$$

$$\tag{13}$$

Note that

$$\left\| \left( \boldsymbol{\Phi}_{m,D} \mathbf{W}_D \boldsymbol{\Phi}_{m,D}^T + \frac{\lambda}{2}I \right)^{-1/2} S_{m,D}^* \right\|$$

$$\leq \left\| \left( \boldsymbol{\Phi}_{m,D} \mathbf{W}_D \boldsymbol{\Phi}_{m,D}^T + \frac{\lambda}{2}I \right)^{-1/2} \boldsymbol{\Phi}_{m,D} \mathbf{W}_D \boldsymbol{\Phi}_{m,D}^T \left( \boldsymbol{\Phi}_{m,D} \mathbf{W}_D \boldsymbol{\Phi}_{m,D}^T + \frac{\lambda}{2}I \right)^{-1/2} \right\|^{1/2} \leq 1$$

and

$$\left\| \left( \boldsymbol{\Phi}_{m,D} \mathbf{W}_D \boldsymbol{\Phi}_{m,D}^T + \frac{\lambda}{2}I \right)^{-1/2} S_m^* \right\|$$

$$= \left\| \left( \boldsymbol{\Phi}_{m,D} \mathbf{W}_D \boldsymbol{\Phi}_{m,D}^T + \frac{\lambda}{2}I \right)^{-1/2} \left( S_m^* S_m + \frac{\lambda}{2}I \right)^{1/2} \left( S_m^* S_m + \frac{\lambda}{2}I \right)^{-1/2} S_m^* \right\|$$

$$\leq \left\| \left( \boldsymbol{\Phi}_{m,D} \mathbf{W}_D \boldsymbol{\Phi}_{m,D}^T + \frac{\lambda}{2}I \right)^{-1/2} \left( S_m^* S_m + \frac{\lambda}{2}I \right)^{1/2} \right\| \left\| \left( S_m^* S_m + \frac{\lambda}{2}I \right)^{-1/2} S_m^* \right\|,$$

since $\left\| \left( S_m^* S_m + \frac{\lambda}{2} I \right)^{-1/2} S_m^* \right\| = \left\| \left( S_m^* S_m + \frac{\lambda}{2} I \right)^{-1/2} S_m^* S_m \left( S_m^* S_m + \frac{\lambda}{2} I \right)^{-1/2} \right\|^{1/2} \leq 1$. Substituting the above two inequalities into Eq.(13) we have

$$\left\| \mathbf{g}_{m,D,\lambda}^{\diamond} - \mathbf{g}_{m,\lambda} \right\| \leq \frac{1}{\sqrt{\lambda}} \left( 1 + \left\| \left( \boldsymbol{\Phi}_{m,D} \mathbf{W}_D \boldsymbol{\Phi}_{m,D}^T + \frac{\lambda}{2} I \right)^{-1/2} \left( S_m^* S_m + \frac{\lambda}{2} I \right)^{1/2} \right\| \right) \| f_{m,\lambda} - f_\rho \|_K .$$
$$(14)$$

Combining Eq.(11) and Eq.(14), we finish this proof. $\qquad\square$

**Lemma 2.** *We have*

$$\| f_{m,D,\lambda} - f_{m,\lambda} \|_K \leq \left\| \left( \boldsymbol{\Phi}_{m,D} \mathbf{W}_D \boldsymbol{\Phi}_{m,D}^T + \frac{\lambda}{2} I \right)^{-1/2} \left( S_m^* S_m + \frac{\lambda}{2} I \right)^{1/2} \right\|^2$$

$$* \left( \left\| \left( S_m^* S_m + \frac{\lambda}{2} I \right)^{-1/2} (\boldsymbol{\Phi}_{m,D} \mathbf{W}_D \bar{\mathbf{y}}_D - S_m^* \mathbf{W}_D f_\rho) \right\| + \left\| \left( S_m^* S_m + \frac{\lambda}{2} I \right)^{-1/2} (S_m^* \mathbf{W}_D f_\rho - S_{m,D}^* \mathbf{W}_D f_\rho) \right\| \right)$$

$$+ \left( \left\| \left( \boldsymbol{\Phi}_{m,D} \mathbf{W}_D \boldsymbol{\Phi}_{m,D}^T + \frac{\lambda}{2} I \right)^{-1/2} \left( S_m^* S_m + \frac{\lambda}{2} I \right)^{1/2} \right\| + \left\| \left( \boldsymbol{\Phi}_{m,D} \mathbf{W}_D \boldsymbol{\Phi}_{m,D}^T + \frac{\lambda}{2} I \right)^{-1/2} \left( S_m^* S_m + \frac{\lambda}{2} I \right)^{1/2} \right\|^2 \right)$$

$$* \| f_{m,\lambda} - f_\rho \|_K .$$

*Proof.* Note that

$$\| f_{m,D,\lambda} - f_{m,\lambda} \|_K \leq \left\| f_{m,D,\lambda} - f_{m,D,\lambda}^{\diamond} \right\|_K + \left\| f_{m,D,\lambda}^{\diamond} - f_{m,\lambda} \right\|_K .$$

According to $f_{m,D,\lambda} - f_{m,D,\lambda}^{\diamond} = S_m \left( \mathbf{g}_{m,D,\lambda} - \mathbf{g}_{m,D,\lambda}^{\diamond} \right)$, by Eq.(10), we have

$$f_{m,D,\lambda} - f_{m,D,\lambda}^{\diamond} = S_m \left( \mathbf{g}_{m,D,\lambda} - \mathbf{g}_{m,D,\lambda}^{\diamond} \right)$$
$$= S_m \left( S_m^* S_m + \frac{\lambda}{2} I \right)^{-1/2} \left( S_m^* S_m + \frac{\lambda}{2} I \right)^{1/2} \left( \boldsymbol{\Phi}_{m,D} \mathbf{W}_D \boldsymbol{\Phi}_{m,D}^T + \frac{\lambda}{2} I \right)^{-1} \left( \boldsymbol{\Phi}_{m,D} \mathbf{W}_D \boldsymbol{\Phi}_{m,D}^T + \frac{\lambda}{2} I \right)^{-1/2}$$
$$* \left( S_m^* S_m + \frac{\lambda}{2} I \right)^{1/2} \left( S_m^* S_m + \frac{\lambda}{2} I \right)^{-1/2} (\boldsymbol{\Phi}_{m,D} \mathbf{W}_D \bar{\mathbf{y}}_D - S_m^* \mathbf{W}_D f_\rho + S_m^* \mathbf{W}_D f_\rho - S_{m,D}^* \mathbf{W}_D f_\rho) .$$
$$(15)$$

Note that

$$\left\| S_m \left( S_m^* S_m + \frac{\lambda}{2} I \right)^{-1/2} \right\| = \left\| \left( S_m^* S_m + \frac{\lambda}{2} I \right)^{-1/2} S_m^* S_m \left( S_m^* S_m + \frac{\lambda}{2} I \right)^{-1/2} \right\|^{1/2}$$

$$= \left\| \left( S_m^* S_m + \frac{\lambda}{2} I \right)^{-1/2} S_m^* S_m \left( S_m^* S_m + \frac{\lambda}{2} I \right)^{-1/2} \right\|^{1/2} \leq 1.$$

So, by Eq.(15) we have

$$\left\| f_{m,D,\lambda} - f_{m,D,\lambda}^{\diamond} \right\|_K \leq \left\| \left( \boldsymbol{\Phi}_{m,D} \mathbf{W}_D \boldsymbol{\Phi}_{m,D}^T + \frac{\lambda}{2} I \right)^{-1/2} \left( S_m^* S_m + \frac{\lambda}{2} I \right)^{1/2} \right\|^2$$

$$* \left( \left\| \left( S_m^* S_m + \frac{\lambda}{2} I \right)^{-1/2} (\boldsymbol{\Phi}_{m,D} \mathbf{W}_D \bar{\mathbf{y}}_D - S_m^* \mathbf{W}_D f_\rho) \right\| \right. \qquad (16)$$

$$\left. + \left\| \left( S_m^* S_m + \frac{\lambda}{2} I \right)^{-1/2} (S_m^* \mathbf{W}_D f_\rho - S_{m,D}^* \mathbf{W}_D f_\rho) \right\| \right) .$$

Similarly, according to Eq.(12), we have

$$f_{m,D,\lambda}^\diamond - f_{m,\lambda} = S_m \left( \mathbf{g}_{m,D,\lambda}^\diamond - \mathbf{g}_{m,\lambda} \right)$$

$$\leq S_m \left( \mathbf{\Phi}_{m,D} \mathbf{W}_D \mathbf{\Phi}_{m,D}^T + \frac{\lambda}{2} I \right)^{-1} S_{m,D}^* [f_\rho - f_{m,\lambda}] + S_m \left( \mathbf{\Phi}_{m,D} \mathbf{W}_D \mathbf{\Phi}_{m,D}^T + \frac{\lambda}{2} I \right)^{-1} S_m^* [f_{m,\lambda} - f_\rho]$$

$$= S_m \left( S_m^* S_m + \frac{\lambda}{2} I \right)^{-1/2} \left( S_m^* S_m + \frac{\lambda}{2} I \right)^{1/2} \left( \mathbf{\Phi}_{m,D} \mathbf{W}_D \mathbf{\Phi}_{m,D}^T + \frac{\lambda}{2} I \right)^{-1/2} \left( \mathbf{\Phi}_{m,D} \mathbf{W}_D \mathbf{\Phi}_{m,D}^T + \frac{\lambda}{2} I \right)^{-1/2}$$

$$* S_{m,D}^* [f_\rho - f_{m,\lambda}] + S_m \left( S_m^* S_m + \frac{\lambda}{2} I \right)^{-1/2} \left( S_m^* S_m + \frac{\lambda}{2} I \right)^{1/2} \left( \mathbf{\Phi}_{m,D} \mathbf{W}_D \mathbf{\Phi}_{m,D}^T + \frac{\lambda}{2} I \right)^{-1/2}$$

$$* \left( \mathbf{\Phi}_{m,D} \mathbf{W}_D \mathbf{\Phi}_{m,D}^T + \frac{\lambda}{2} I \right)^{-1/2} \left( S_m^* S_m + \frac{\lambda}{2} I \right)^{1/2} \left( S_m^* S_m + \frac{\lambda}{2} I \right)^{-1/2} S_m^* [f_{m,\lambda} - f_\rho].$$

Note that $\left\| S_m \left( S_m^* S_m + \frac{\lambda}{2} I \right)^{-1/2} \right\| = \left\| \left( S_m^* S_m + \frac{\lambda}{2} I \right)^{-1/2} S_m^* S_m \left( S_m^* S_m + \frac{\lambda}{2} I \right)^{-1/2} \right\|^{1/2} \leq 1$,
so we have

$$\begin{aligned}
& \| f_{m,D,\lambda}^\diamond - f_{m,\lambda} \|_K \\
& \leq \left( \left\| \left( \mathbf{\Phi}_{m,D} \mathbf{W}_D \mathbf{\Phi}_{m,D}^T + \frac{\lambda}{2} I \right)^{-1/2} \left( S_m^* S_m + \frac{\lambda}{2} I \right)^{1/2} \right\| \right. \\
& \quad \left. + \left\| \left( \mathbf{\Phi}_{m,D} \mathbf{W}_D \mathbf{\Phi}_{m,D}^T + \frac{\lambda}{2} I \right)^{-1/2} \left( S_m^* S_m + \frac{\lambda}{2} I \right)^{1/2} \right\|^2 \right) \| f_{m,\lambda} - f_\rho \|_K .
\end{aligned} \tag{17}$$

Combining Eq.(16) and Eq.(17), we finish this proof.

$\square$

**Lemma 3.** *For $\delta \in (0,1]$, with probability at least $1 - \delta$, we have*

$$\left\| \left( S_m^* S_m + \frac{\lambda}{2} I \right)^{-1/2} \left( \mathbf{\Phi}_{m,D} \mathbf{W}_D \bar{\mathbf{y}}_D - S_m^* \mathbf{W}_D f_\rho \right) \right\| = \mathcal{O} \left( \left( \frac{1}{\sqrt{\lambda}|D|} + \sqrt{\frac{\mathcal{N}_m(\lambda)}{|D|}} \right) \log \frac{1}{\delta} \right),$$

*where $\mathcal{N}_m(\lambda) = \mathrm{Tr} \left( \left( L_m + \frac{\lambda}{2} I \right)^{-1} L_m \right)$, $L_m$ is the integral operator associated with the approximate kernel function $K_m$, $(L_m f)(\mathbf{x}) = \int_\mathcal{X} K_m(\mathbf{x}, \mathbf{x}') f(\mathbf{x}') d\rho_\mathcal{X}(\mathbf{x}')$.*

*Proof.* We have

$$\left\| \left( S_m^* S_m + \frac{\lambda}{2} I \right)^{-1/2} \left( \mathbf{\Phi}_{m,D} \mathbf{W}_D \bar{\mathbf{y}}_D - S_m^* \mathbf{W}_D f_\rho \right) \right\| \leq \left\| \left( S_m^* S_m + \frac{\lambda}{2} I \right)^{-1/2} \left( \mathbf{\Phi}_{m,D} \bar{\mathbf{y}}_D - S_m^* f_\rho \right) \right\|.$$

According to Lemma 6 in Rudi & Rosasco (2017), we know, with probability at least $1 - \delta$,

$$\left\| \left( S_m^* S_m + \frac{\lambda}{2} I \right)^{-1/2} \left( \mathbf{\Phi}_{m,D} \bar{\mathbf{y}}_D - S_m^* f_\rho \right) \right\| = \mathcal{O} \left( \left( \frac{1}{\sqrt{\lambda}|D|} + \sqrt{\frac{\mathcal{N}_m(\lambda)}{|D|}} \right) \log \frac{1}{\delta} \right).$$

where $\mathcal{N}_m(\lambda) = \mathrm{Tr} \left( \left( L_m + \frac{\lambda}{2} I \right)^{-1} L_m \right)$, $L_m$ is the integral operator associated with the approximate kernel function $K_m$, $(L_m f)(\mathbf{x}) = \int_\mathcal{X} K_m(\mathbf{x}, \mathbf{x}') f(\mathbf{x}') d\rho_\mathcal{X}(\mathbf{x}')$. Thus, we complete this proof.

$\square$

**Lemma 4.** *For $\delta \in (0,1]$, with probability at least $1 - \delta$, we have*

$$\left\| \left( S_m^* S_m + \frac{\lambda}{2} I \right)^{-1/2} \left( S_m^* \mathbf{W}_D f_\rho - S_{m,D}^* \mathbf{W}_D f_\rho \right) \right\| \leq \frac{\tau \zeta \log \frac{1}{\delta}}{|D|\sqrt{\lambda}} + 2\zeta \sqrt{\frac{\mathcal{N}_m(\lambda)}{|D|}},$$

*where $\mathcal{N}_m(\lambda) = \mathrm{Tr} \left( \left( L_m + \frac{\lambda}{2} I \right)^{-1} L_m \right)$.*

*Proof.* We have

$$\left\| \left( S_m^* S_m + \frac{\lambda}{2} I \right)^{-1/2} (S_m^* \mathbf{W}_D f_\rho - S_{m,D}^* \mathbf{W}_D f_\rho) \right\| \leq \left\| \left( S_m^* S_m + \frac{\lambda}{2} I \right)^{-1/2} (S_m^* f_\rho - S_{m,D}^* f_\rho) \right\|.$$

According to Proposition 5 in Liu et al. (2021), with probability at least $1 - \delta$, we have

$$\left\| \left( S_m^* S_m + \frac{\lambda}{2} I \right)^{-1/2} (S_m^* f_\rho - S_{m,D}^* f_\rho) \right\| \leq \frac{\tau \zeta \log \frac{1}{\delta}}{|D| \sqrt{\lambda}} + 2\zeta \sqrt{\frac{\mathcal{N}_m(\lambda)}{|D|}},$$

where $\mathcal{N}_m(\lambda) = \mathrm{Tr} \left( \left( L_m + \frac{\lambda}{2} I \right)^{-1} L_m \right)$. Combining them, we complete this proof. $\square$

**Lemma 5.** *For any $\delta > 0$, with probability at least $1 - \delta$, we have*

$$\left\| \left( S_m^* S_m + \frac{\lambda}{2} I \right)^{-1} (S_m^* S_m - \mathbf{\Phi}_{m,D} \mathbf{W}_D \mathbf{\Phi}_{m,D}^T) \right\|$$

$$= \left\| \left( S_m^* S_m + \frac{\lambda}{2} I \right)^{-1/2} (S_m^* S_m - \mathbf{\Phi}_{m,D} \mathbf{W}_D \mathbf{\Phi}_{m,D}^T) \left( S_m^* S_m + \frac{\lambda}{2} I \right)^{-1/2} \right\|$$

$$\leq \frac{2 \log^2(2/\delta) \left( 2\tau^2 \lambda^{-1} + 1 \right)}{|D|} + \sqrt{\frac{2 \log(2/\delta) \left( 2\tau^2 \lambda^{-1} + 1 \right)}{|D|}}.$$

*Proof.* Since $S_m^* S_m$ is self-adjoint operator, so we have

$$\left\| \left( S_m^* S_m + \frac{\lambda}{2} I \right)^{-1} (S_m^* S_m - \mathbf{\Phi}_{m,D} \mathbf{W}_D \mathbf{\Phi}_{m,D}^T) \right\|$$

$$= \left\| \left( S_m^* S_m + \frac{\lambda}{2} I \right)^{-1/2} (S_m^* S_m - \mathbf{\Phi}_{m,D} \mathbf{W}_D \mathbf{\Phi}_{m,D}^T) \left( S_m^* S_m + \frac{\lambda}{2} I \right)^{-1/2} \right\|.$$

According to Proposition 1 with $\zeta_i = \phi_m(\mathbf{x}_i)$, we can obtain

$$\left\| \left( S_m^* S_m + \frac{\lambda}{2} I \right)^{-1} (S_m^* S_m - \mathbf{\Phi}_{m,D} \mathbf{W}_D \mathbf{\Phi}_{m,D}^T) \right\| \leq \frac{2 \log^2(2/\delta) \left( \mathcal{N}_\infty(\lambda) + 1 \right)}{|D|} + \sqrt{\frac{2 \log(2/\delta) \left( \mathcal{N}_\infty(\lambda) + 1 \right)}{|D|}},$$

where

$$\mathcal{N}_\infty(\lambda) = \sup_{\omega \in \Omega} \left\| \left( \tilde{L}_K + \frac{\lambda}{2} I \right)^{-1/2} \psi(\cdot, \boldsymbol{\omega}) \right\|_K^2 \leq 2\tau^2 \lambda^{-1},$$

$\tilde{L}_K f = \int_{\mathcal{X}} K(\mathbf{x}, \cdot) f(\mathbf{x}) d\rho_{\mathcal{X}}$ (Rudi & Rosasco, 2017), $c_1$ and $c_2$ are two constants.
Therefore, we have

$$\left\| \left( S_m^* S_m + \frac{\lambda}{2} I \right)^{-1} (S_m^* S_m - \mathbf{\Phi}_{m,D} \mathbf{W}_D \mathbf{\Phi}_{m,D}^T) \right\| \leq \frac{2 \log^2(2/\delta) \left( 2\tau^2 \lambda^{-1} + 1 \right)}{|D|} + \sqrt{\frac{2 \log(2/\delta) \left( 2\tau^2 \lambda^{-1} + 1 \right)}{|D|}}.$$

$\square$

**Lemma 6.** *We have*

$$\left\| \bar{\mathbf{g}}_{m,D,\lambda}^0 - \mathbf{g}_{m,D,\lambda} \right\|$$

$$\leq \sum_{j=1}^p \frac{|D_j|^2}{\sum_{k=1}^p |D_k|^2} \left\| \left( \mathbf{\Phi}_{m,D} \mathbf{W}_D \mathbf{\Phi}_{m,D}^T + \frac{\lambda}{2} I \right)^{-1/2} \left( S_m^* S_m + \frac{\lambda}{2} I \right)^{1/2} \right\|^2$$

$$* \left( \left\| \left( S_m^* S_m + \frac{\lambda}{2} I \right)^{-1/2} \left( S_m^* S_m - \mathbf{\Phi}_{m,D} \mathbf{W}_D \mathbf{\Phi}_{m,D}^T \right) \left( S_m^* S_m + \frac{\lambda}{2} I \right)^{-1/2} \right\| \right. \tag{18}$$

$$\left. + \left\| \left( S_m^* S_m + \frac{\lambda}{2} I \right)^{-1/2} \left( S_m^* S_m - \mathbf{\Phi}_{m,D_j} \mathbf{W}_{D_j} \mathbf{\Phi}_{m,D_j}^T \right) \left( S_m^* S_m + \frac{\lambda}{2} I \right)^{-1/2} \right\| \right)$$

$$* \left\| \mathbf{g}_{m,D_j,\lambda} - \mathbf{g}_{m,\lambda} \right\|.$$

*Proof.* Note that $\mathbf{g}_{m,D,\lambda} = (\mathbf{\Phi}_{m,D} \mathbf{W}_D \mathbf{\Phi}_{m,D}^T + \frac{\lambda}{2} I)^{-1} \mathbf{\Phi}_{m,D} \mathbf{W}_D \bar{\mathbf{y}}_D$. Thus we have

$$\bar{\mathbf{g}}_{m,D,\lambda}^0 - \mathbf{g}_{m,D,\lambda}$$

$$= \sum_{j=1}^p \frac{|D_j|^2}{\sum_{k=1}^p |D_k|^2} (\mathbf{\Phi}_{m,D_j} \mathbf{W}_{D_j} \mathbf{\Phi}_{m,D_j}^T + \frac{\lambda}{2} I)^{-1} \mathbf{\Phi}_{m,D_j} \mathbf{W}_{D_j} \bar{\mathbf{y}}_{D_j}$$

$$\quad - (\mathbf{\Phi}_{m,D} \mathbf{W}_D \mathbf{\Phi}_{m,D}^T + \frac{\lambda}{2} I)^{-1} \mathbf{\Phi}_{m,D} \mathbf{W}_D \bar{\mathbf{y}}_D$$

$$= \sum_{j=1}^p \frac{|D_j|^2}{\sum_{k=1}^p |D_k|^2} \left( \left( \mathbf{\Phi}_{m,D_j} \mathbf{W}_{D_j} \mathbf{\Phi}_{m,D_j}^T + \frac{\lambda}{2} I \right)^{-1} - \left( \mathbf{\Phi}_{m,D} \mathbf{W}_D \mathbf{\Phi}_{m,D}^T + \frac{\lambda}{2} I \right)^{-1} \right) \mathbf{\Phi}_{m,D_j} \mathbf{W}_{D_j} \bar{\mathbf{y}}_{D_j}$$

$$= \sum_{j=1}^p \frac{|D_j|^2}{\sum_{k=1}^p |D_k|^2} \left( \mathbf{\Phi}_{m,D} \mathbf{W}_D \mathbf{\Phi}_{m,D}^T + \frac{\lambda}{2} I \right)^{-1} \left( \mathbf{\Phi}_{m,D} \mathbf{W}_D \mathbf{\Phi}_{m,D}^T - \mathbf{\Phi}_{m,D_j} \mathbf{W}_{D_j} \mathbf{\Phi}_{m,D_j}^T \right)$$

$$\quad * \left( \mathbf{\Phi}_{m,D_j} \mathbf{W}_{D_j} \mathbf{\Phi}_{m,D_j}^T + \frac{\lambda}{2} I \right)^{-1} \mathbf{\Phi}_{m,D_j} \mathbf{W}_{D_j} \bar{\mathbf{y}}_{D_j}$$

$$= \sum_{j=1}^p \frac{|D_j|^2}{\sum_{k=1}^p |D_k|^2} \left( \mathbf{\Phi}_{m,D} \mathbf{W}_D \mathbf{\Phi}_{m,D}^T + \frac{\lambda}{2} I \right)^{-1} \left( \mathbf{\Phi}_{m,D} \mathbf{W}_D \mathbf{\Phi}_{m,D}^T - \mathbf{\Phi}_{m,D_j} \mathbf{W}_{D_j} \mathbf{\Phi}_{m,D_j}^T \right) \mathbf{g}_{m,D_j,\lambda}$$

$$\tag{19}$$

By introducing $S_m^* S_m$ term, we can convert the above formula into

$$\bar{\mathbf{g}}_{m,D,\lambda}^0 - \mathbf{g}_{m,D,\lambda}$$

$$= \sum_{j=1}^p \frac{|D_j|^2}{\sum_{k=1}^p |D_k|^2} \left( \mathbf{\Phi}_{m,D} \mathbf{W}_D \mathbf{\Phi}_{m,D}^T + \frac{\lambda}{2} I \right)^{-1} \left( \mathbf{\Phi}_{m,D} \mathbf{W}_D \mathbf{\Phi}_{m,D}^T - S_m^* S_m \right) \mathbf{g}_{m,D_j,\lambda}$$

$$\quad + \sum_{j=1}^p \frac{|D_j|^2}{\sum_{k=1}^p |D_k|^2} \left( \mathbf{\Phi}_{m,D} \mathbf{W}_D \mathbf{\Phi}_{m,D}^T + \frac{\lambda}{2} I \right)^{-1} \left( S_m^* S_m - \mathbf{\Phi}_{m,D_j} \mathbf{W}_{D_j} \mathbf{\Phi}_{m,D_j}^T \right) \mathbf{g}_{m,D_j,\lambda}$$

$$= \sum_{j=1}^p \frac{|D_j|^2}{\sum_{k=1}^p |D_k|^2} \left( \mathbf{\Phi}_{m,D} \mathbf{W}_D \mathbf{\Phi}_{m,D}^T + \frac{\lambda}{2} I \right)^{-1} \left( \mathbf{\Phi}_{m,D} \mathbf{W}_D \mathbf{\Phi}_{m,D}^T - S_m^* S_m \right) \left( \mathbf{g}_{m,D_j,\lambda} - \mathbf{g}_{m,\lambda} \right)$$

$$\quad + \sum_{j=1}^p \frac{|D_j|^2}{\sum_{k=1}^p |D_k|^2} \left( \mathbf{\Phi}_{m,D} \mathbf{W}_D \mathbf{\Phi}_{m,D}^T + \frac{\lambda}{2} I \right)^{-1} \left( \mathbf{\Phi}_{m,D} \mathbf{W}_D \mathbf{\Phi}_{m,D}^T - S_m^* S_m \right) \mathbf{g}_{m,\lambda}$$

$$\quad + \sum_{j=1}^p \frac{|D_j|^2}{\sum_{k=1}^p |D_k|^2} \left( \mathbf{\Phi}_{m,D} \mathbf{W}_D \mathbf{\Phi}_{m,D}^T + \frac{\lambda}{2} I \right)^{-1} \left( S_m^* S_m - \mathbf{\Phi}_{m,D_j} \mathbf{W}_{D_j} \mathbf{\Phi}_{m,D_j}^T \right) \mathbf{g}_{m,D_j,\lambda}.$$

$$\tag{20}$$

So we have

$$
\bar{\mathbf{g}}^0_{m,D,\lambda} - \mathbf{g}_{m,D,\lambda}
$$

$$
= \underbrace{\sum_{j=1}^{p} \frac{|D_j|^2}{\sum_{k=1}^{p}|D_k|^2} \left( \mathbf{\Phi}_{m,D}\mathbf{W}_D\mathbf{\Phi}_{m,D}^T + \frac{\lambda}{2}I \right)^{-1} \left( \mathbf{\Phi}_{m,D}\mathbf{W}_D\mathbf{\Phi}_{m,D}^T - S_m^*S_m \right) \left( \mathbf{g}_{m,D_j,\lambda} - \mathbf{g}_{m,\lambda} \right)}_{\textbf{Term-A}}
$$

$$
+ \underbrace{\sum_{j=1}^{p} \frac{|D_j|^2}{\sum_{k=1}^{p}|D_k|^2} \left( \mathbf{\Phi}_{m,D}\mathbf{W}_D\mathbf{\Phi}_{m,D}^T + \frac{\lambda}{2}I \right)^{-1} \left( S_m^*S_m - \mathbf{\Phi}_{m,D_j}\mathbf{W}_{D_j}\mathbf{\Phi}_{m,D_j}^T \right) \left( \mathbf{g}_{m,D_j,\lambda} - \mathbf{g}_{m,\lambda} \right)}_{\textbf{Term-B}}.
$$

$$(21)$$

Note that

$$
\textbf{Term-A} = \sum_{j=1}^{p} \frac{|D_j|^2}{\sum_{k=1}^{p}|D_k|^2} \left( \mathbf{\Phi}_{m,D}\mathbf{W}_D\mathbf{\Phi}_{m,D}^T + \frac{\lambda}{2}I \right)^{-1} \left( S_m^*S_m + \frac{\lambda}{2}I \right) \left( S_m^*S_m + \frac{\lambda}{2}I \right)^{-1}
$$

$$
* \left( \mathbf{\Phi}_{m,D}\mathbf{W}_D\mathbf{\Phi}_{m,D}^T - S_m^*S_m \right) \left( \mathbf{g}_{m,D_j,\lambda} - \mathbf{g}_{m,\lambda} \right)
$$

and

$$
\textbf{Term-B} = \sum_{j=1}^{p} \frac{|D_j|^2}{\sum_{k=1}^{p}|D_k|^2} \left( \mathbf{\Phi}_{m,D}\mathbf{W}_D\mathbf{\Phi}_{m,D}^T + \frac{\lambda}{2}I \right)^{-1} \left( S_m^*S_m + \frac{\lambda}{2}I \right)
$$

$$
* \left( S_m^*S_m + \frac{\lambda}{2}I \right)^{-1} \left( S_m^*S_m - \mathbf{\Phi}_{m,D_j}\mathbf{W}_{D_j}\mathbf{\Phi}_{m,D_j}^T \right) \left( \mathbf{g}_{m,D_j,\lambda} - \mathbf{g}_{m,\lambda} \right).
$$

Substituting the above equations into Eq.(21), we have

$$
\left\| \bar{\mathbf{g}}^0_{m,D,\lambda} - \mathbf{g}_{m,D,\lambda} \right\|
$$

$$
\leq \sum_{j=1}^{p} \frac{|D_j|^2}{\sum_{k=1}^{p}|D_k|^2} \left\| \left( \mathbf{\Phi}_{m,D}\mathbf{W}_D\mathbf{\Phi}_{m,D}^T + \frac{\lambda}{2}I \right)^{-1/2} \left( S_m^*S_m + \frac{\lambda}{2}I \right)^{1/2} \right\|^2
$$

$$
* \left( \left\| \left( S_m^*S_m + \frac{\lambda}{2}I \right)^{-1/2} \left( S_m^*S_m - \mathbf{\Phi}_{m,D}\mathbf{W}_D\mathbf{\Phi}_{m,D}^T \right) \left( S_m^*S_m + \frac{\lambda}{2}I \right)^{-1/2} \right\| \right.
$$

$$
\left. + \left\| \left( S_m^*S_m + \frac{\lambda}{2}I \right)^{-1/2} \left( S_m^*S_m - \mathbf{\Phi}_{m,D_j}\mathbf{W}_{D_j}\mathbf{\Phi}_{m,D_j}^T \right) \left( S_m^*S_m + \frac{\lambda}{2}I \right)^{-1/2} \right\| \right)
$$

$$
* \left\| \mathbf{g}_{m,D_j,\lambda} - \mathbf{g}_{m,\lambda} \right\|.
$$

Here, we complete this proof. $\qquad \square$

**Lemma 7.** *We have*

$$
\left\| \bar{f}^0_{m,D,\lambda} - f_{m,D,\lambda} \right\|
$$

$$
\leq \sum_{j=1}^{p} \frac{|D_j|^2}{\sum_{k=1}^{p}|D_k|^2} \left\| \left( \mathbf{\Phi}_{m,D}\mathbf{W}_D\mathbf{\Phi}_{m,D}^T + \frac{\lambda}{2}I \right)^{-1/2} \left( S_m^*S_m + \frac{\lambda}{2}I \right)^{1/2} \right\|^2
$$

$$
* \left( \left\| \left( S_m^*S_m + \frac{\lambda}{2}I \right)^{-1/2} \left( S_m^*S_m - \mathbf{\Phi}_{m,D}\mathbf{W}_D\mathbf{\Phi}_{m,D}^T \right) \left( S_m^*S_m + \frac{\lambda}{2}I \right)^{-1/2} \right\| \right. \qquad (22)
$$

$$
\left. + \left\| \left( S_m^*S_m + \frac{\lambda}{2}I \right)^{-1/2} \left( S_m^*S_m - \mathbf{\Phi}_{m,D_j}\mathbf{W}_{D_j}\mathbf{\Phi}_{m,D_j}^T \right) \left( S_m^*S_m + \frac{\lambda}{2}I \right)^{-1/2} \right\| \right)
$$

$$
* \left( \left\| f_{m,D_j,\lambda} - f_{m,\lambda} \right\|_K + \sqrt{\lambda} \left\| \mathbf{g}_{m,D_j,\lambda} - \mathbf{g}_{m,\lambda} \right\| \right).
$$

*Proof.* Note that $S_m \left( \bar{\mathbf{g}}^0_{m,D,\lambda} - \mathbf{g}_{m,D,\lambda} \right) = \bar{f}^0_{m,D,\lambda} - f_{m,D,\lambda}$.

According to Eq.(21), we have

$$
\bar{f}^0_{m,D,\lambda} - f_{m,D,\lambda}
$$
$$
= \sum_{j=1}^{p} \frac{|D_j|^2}{\sum_{k=1}^{p} |D_k|^2} \underbrace{S_m \left( \mathbf{\Phi}_{m,D} \mathbf{W}_D \mathbf{\Phi}^T_{m,D} + \frac{\lambda}{2} I \right)^{-1} \left( \mathbf{\Phi}_{m,D} \mathbf{W}_D \mathbf{\Phi}^T_{m,D} - S^*_m S_m \right) \left( \mathbf{g}_{m,D_j,\lambda} - \mathbf{g}_{m,\lambda} \right)}_{\textbf{Term-A}}
$$
$$
+ \sum_{j=1}^{p} \frac{|D_j|^2}{\sum_{k=1}^{p} |D_k|^2} \underbrace{S_m \left( \mathbf{\Phi}_{m,D} \mathbf{W}_D \mathbf{\Phi}^T_{m,D} + \frac{\lambda}{2} I \right)^{-1} \left( S^*_m S_m - \mathbf{\Phi}_{m,D_j} \mathbf{W}_{D_j} \mathbf{\Phi}^T_{m,D_j} \right) \left( \mathbf{g}_{m,D_j,\lambda} - \mathbf{g}_{m,\lambda} \right)}_{\textbf{Term-B}} .
$$

$$(23)$$

Note that

**Term-A**
$$
= S_m \left( S^*_m S_m + \frac{\lambda}{2} I \right)^{-1/2} \left( S^*_m S_m + \frac{\lambda}{2} I \right)^{1/2} \left( \mathbf{\Phi}_{m,D} \mathbf{W}_D \mathbf{\Phi}^T_{m,D} + \frac{\lambda}{2} I \right)^{-1/2}
$$
$$
* \left( \mathbf{\Phi}_{m,D} \mathbf{W}_D \mathbf{\Phi}^T_{m,D} + \frac{\lambda}{2} I \right)^{-1/2} \left( S^*_m S_m + \frac{\lambda}{2} I \right)^{1/2}
$$
$$
* \left( S^*_m S_m + \frac{\lambda}{2} I \right)^{-1/2} \left( \mathbf{\Phi}_{m,D} \mathbf{W}_D \mathbf{\Phi}^T_{m,D} - S^*_m S_m \right) \left( S^*_m S_m + \frac{\lambda}{2} I \right)^{-1/2}
$$
$$
* \left( S^*_m S_m + \frac{\lambda}{2} I \right)^{-1/2} \left( S^*_m S_m + \frac{\lambda}{2} I \right) \left( \mathbf{g}_{m,D_j,\lambda} - \mathbf{g}_{m,\lambda} \right) .
$$

So, we have

$$
\| \textbf{Term-A} \|_K
$$
$$
\leq \left\| \left( \mathbf{\Phi}_{m,D} \mathbf{W}_D \mathbf{\Phi}^T_{m,D} + \frac{\lambda}{2} I \right)^{-1/2} \left( S^*_m S_m + \frac{\lambda}{2} I \right)^{1/2} \right\|^2
$$
$$
* \left\| \left( S^*_m S_m + \frac{\lambda}{2} I \right)^{-1/2} \left( S^*_m S_m - \mathbf{\Phi}_{m,D} \mathbf{W}_D \mathbf{\Phi}^T_{m,D} \right) \left( S^*_m S_m + \frac{\lambda}{2} I \right)^{-1/2} \right\|
$$
$$
* \left\| S_m \left( S^*_m S_m + \frac{\lambda}{2} I \right)^{-1/2} \right\| \left\| \left( S^*_m S_m + \frac{\lambda}{2} I \right)^{-1/2} \left( S^*_m S_m + \frac{\lambda}{2} I \right) \left( \mathbf{g}_{m,D_j,\lambda} - \mathbf{g}_{m,\lambda} \right) \right\|
$$
$$
\leq \left\| \left( \mathbf{\Phi}_{m,D} \mathbf{W}_D \mathbf{\Phi}^T_{m,D} + \frac{\lambda}{2} I \right)^{-1/2} \left( S^*_m S_m + \frac{\lambda}{2} I \right)^{1/2} \right\|^2
$$
$$
* \left\| \left( S^*_m S_m + \frac{\lambda}{2} I \right)^{-1/2} \left( S^*_m S_m - \mathbf{\Phi}_{m,D} \mathbf{W}_D \mathbf{\Phi}^T_{m,D} \right) \left( S^*_m S_m + \frac{\lambda}{2} I \right)^{-1/2} \right\|
$$
$$
* \left\| \left( S^*_m S_m + \frac{\lambda}{2} I \right)^{-1/2} \left( S^*_m S_m + \frac{\lambda}{2} I \right) \left( \mathbf{g}_{m,D_j,\lambda} - \mathbf{g}_{m,\lambda} \right) \right\| .
$$

Since $\left\| S_m \left( S^*_m S_m + \frac{\lambda}{2} I \right)^{-1/2} \right\| = \left\| \left( S^*_m S_m + \frac{\lambda}{2} I \right)^{-1/2} S^*_m S_m \left( S^*_m S_m + \frac{\lambda}{2} I \right)^{-1/2} \right\|^{1/2} \leq 1.$

So, we have

$\|\textbf{Term-A}\|_K$

$$\leq \left\| \left( \boldsymbol{\Phi}_{m,D} \mathbf{W}_D \boldsymbol{\Phi}_{m,D}^T + \frac{\lambda}{2} I \right)^{-1/2} \left( S_m^* S_m + \frac{\lambda}{2} I \right)^{1/2} \right\|^2$$

$$* \left\| \left( S_m^* S_m + \frac{\lambda}{2} I \right)^{-1/2} \left( S_m^* S_m - \boldsymbol{\Phi}_{m,D} \mathbf{W}_D \boldsymbol{\Phi}_{m,D}^T \right) \left( S_m^* S_m + \frac{\lambda}{2} I \right)^{-1/2} \right\|$$

$$* \left\| \left( S_m^* S_m + \frac{\lambda}{2} I \right)^{-1/2} \left( S_m^* S_m + \frac{\lambda}{2} I \right) \left( \mathbf{g}_{m,D_j,\lambda} - \mathbf{g}_{m,\lambda} \right) \right\|$$

$$= \left\| \left( \boldsymbol{\Phi}_{m,D} \mathbf{W}_D \boldsymbol{\Phi}_{m,D}^T + \frac{\lambda}{2} I \right)^{-1/2} \left( S_m^* S_m + \frac{\lambda}{2} I \right)^{1/2} \right\|^2$$

$$* \left\| \left( S_m^* S_m + \frac{\lambda}{2} I \right)^{-1/2} \left( S_m^* S_m - \boldsymbol{\Phi}_{m,D} \mathbf{W}_D \boldsymbol{\Phi}_{m,D}^T \right) \left( S_m^* S_m + \frac{\lambda}{2} I \right)^{-1/2} \right\|$$

$$* \left\| \left( S_m^* S_m + \frac{\lambda}{2} I \right)^{-1/2} \left( S_m^* S_m + \frac{\lambda}{2} I \right) \left( \mathbf{g}_{m,D_j,\lambda} - \mathbf{g}_{m,\lambda} \right) \right\|$$

$$\leq \left\| \left( \boldsymbol{\Phi}_{m,D} \mathbf{W}_D \boldsymbol{\Phi}_{m,D}^T + \frac{\lambda}{2} I \right)^{-1/2} \left( S_m^* S_m + \frac{\lambda}{2} I \right)^{1/2} \right\|^2$$

$$* \left\| \left( S_m^* S_m + \frac{\lambda}{2} I \right)^{-1/2} \left( S_m^* S_m - \boldsymbol{\Phi}_{m,D} \mathbf{W}_D \boldsymbol{\Phi}_{m,D}^T \right) \left( S_m^* S_m + \frac{\lambda}{2} I \right)^{-1/2} \right\|$$

$$* \left\| \left( S_m^* S_m + \frac{\lambda}{2} I \right)^{-1/2} S_m^* S_m \left( \mathbf{g}_{m,D_j,\lambda} - \mathbf{g}_{m,\lambda} \right) \right\| + \frac{\lambda}{2} \left\| \left( \boldsymbol{\Phi}_{m,D} \mathbf{W}_D \boldsymbol{\Phi}_{m,D}^T + \frac{\lambda}{2} I \right)^{-1/2} \left( S_m^* S_m + \frac{\lambda}{2} I \right)^{1/2} \right\|^2$$

$$* \left\| \left( S_m^* S_m + \frac{\lambda}{2} I \right)^{-1/2} \left( S_m^* S_m - \boldsymbol{\Phi}_{m,D} \mathbf{W}_D \boldsymbol{\Phi}_{m,D}^T \right) \left( S_m^* S_m + \frac{\lambda}{2} I \right)^{-1/2} \right\|$$

$$* \left\| \left( S_m^* S_m + \frac{\lambda}{2} I \right)^{-1/2} \left( \mathbf{g}_{m,D_j,\lambda} - \mathbf{g}_{m,\lambda} \right) \right\|$$

$$\leq \left\| \left( \boldsymbol{\Phi}_{m,D} \mathbf{W}_D \boldsymbol{\Phi}_{m,D}^T + \frac{\lambda}{2} I \right)^{-1/2} \left( S_m^* S_m + \frac{\lambda}{2} I \right)^{1/2} \right\|^2$$

$$* \left\| \left( S_m^* S_m + \frac{\lambda}{2} I \right)^{-1/2} \left( S_m^* S_m - \boldsymbol{\Phi}_{m,D} \mathbf{W}_D \boldsymbol{\Phi}_{m,D}^T \right) \left( S_m^* S_m + \frac{\lambda}{2} I \right)^{-1/2} \right\|$$

$$* \left\| \left( S_m^* S_m + \frac{\lambda}{2} I \right)^{-1/2} S_m^* \right\| \left\| f_{m,D_j,\lambda} - f_{m,D,\lambda} \right\|_K + \sqrt{\lambda} \left\| \left( \boldsymbol{\Phi}_{m,D} \mathbf{W}_D \boldsymbol{\Phi}_{m,D}^T + \frac{\lambda}{2} I \right)^{-1/2} \left( S_m^* S_m + \frac{\lambda}{2} I \right)^{1/2} \right\|^2$$

$$* \left\| \left( S_m^* S_m + \frac{\lambda}{2} I \right)^{-1/2} \left( S_m^* S_m - \boldsymbol{\Phi}_{m,D} \mathbf{W}_D \boldsymbol{\Phi}_{m,D}^T \right) \left( S_m^* S_m + \frac{\lambda}{2} I \right)^{-1/2} \right\|$$

$$* \left\| \left( \mathbf{g}_{m,D_j,\lambda} - \mathbf{g}_{m,\lambda} \right) \right\|$$

$$\leq \left\| \left( \boldsymbol{\Phi}_{m,D} \mathbf{W}_D \boldsymbol{\Phi}_{m,D}^T + \frac{\lambda}{2} I \right)^{-1/2} \left( S_m^* S_m + \frac{\lambda}{2} I \right)^{1/2} \right\|^2$$

$$* \left\| \left( S_m^* S_m + \frac{\lambda}{2} I \right)^{-1/2} \left( S_m^* S_m - \boldsymbol{\Phi}_{m,D} \mathbf{W}_D \boldsymbol{\Phi}_{m,D}^T \right) \left( S_m^* S_m + \frac{\lambda}{2} I \right)^{-1/2} \right\|$$

$$* \left( \left\| f_{m,D_j,\lambda} - f_{m,D,\lambda} \right\|_K + \sqrt{\lambda} \left\| \mathbf{g}_{m,D_j,\lambda} - \mathbf{g}_{m,\lambda} \right\| \right),$$

$$(24)$$

the last inequality uses the fact that

$$\left\| \left(S_m^* S_m + \frac{\lambda}{2} I\right)^{-1/2} S_m^* \right\| = \left\| \left(S_m^* S_m + \frac{\lambda}{2} I\right)^{-1/2} S_m^* S_m \left(S_m^* S_m + \frac{\lambda}{2} I\right)^{-1/2} \right\|^{1/2} \leq 1.$$

Similar as the above process, we can obtain that

$$
\begin{aligned}
&\|\mathbf{Term\text{-}B}\|_K \\
&\leq \left\| \left(\mathbf{\Phi}_{m,D} \mathbf{W}_D \mathbf{\Phi}_{m,D}^T + \frac{\lambda}{2} I\right)^{-1/2} \left(S_m^* S_m + \frac{\lambda}{2} I\right)^{1/2} \right\|^2 \\
&\quad * \left\| \left(S_m^* S_m + \frac{\lambda}{2} I\right)^{-1/2} \left(S_m^* S_m - \mathbf{\Phi}_{m,D_j} \mathbf{W}_{D_j} \mathbf{\Phi}_{m,D_j}^T \right) \left(S_m^* S_m + \frac{\lambda}{2} I\right)^{-1/2} \right\| \\
&\quad * \left( \left\| f_{m,D_j,\lambda} - f_{m,D,\lambda} \right\|_K - \sqrt{\lambda} \left\| \mathbf{g}_{m,D_j,\lambda} - \mathbf{g}_{m,\lambda} \right\| \right).
\end{aligned}
\tag{25}
$$

Combining Eq.(23), Eq.(24), and Eq.(25), we obtain this result. $\qquad \square$

**Lemma 8.** *For $\delta \in (0,1]$ and $\lambda > 0$, when*

$$m = \Omega \left( \lambda^{-2r} \vee \lambda^{-1} \log \frac{1}{\lambda \delta} \right),$$

*with probability at least $1 - \delta$, we have*

$$\|f_{m,\lambda} - f_\lambda\|_K \leq c\lambda^r,$$

*where $c$ is a constant.*

*Proof.* Note that $f_{m,\lambda} = S_m \mathbf{g}_{m,\lambda}$ and $\mathbf{g}_{m,\lambda} = \left(S_m^* \mathbf{W}_D S_m + \frac{\lambda}{2} I\right)^{-1} S_m^* \mathbf{W}_D f_\rho$.

We have $\|f_{m,\lambda} - f_\lambda\|_K = \left\| S_m \left(S_m^* \mathbf{W}_D S_m + \frac{\lambda}{2} I\right)^{-1} S_m^* \mathbf{W}_D f_\rho - f_\lambda \right\|_K \leq \left\| S_m \left(S_m^* S_m + \frac{\lambda}{2} I\right)^{-1} S_m^* f_\rho - \tilde{f}_\lambda \right\|_K$, where $\tilde{f}_\lambda = \arg\min_{f \in \mathcal{H}_K} \{ \int_{\mathcal{X}} (f(\mathbf{x}) - f_\rho(\mathbf{x}))^2 d\rho_{\mathcal{X}}(\mathbf{x}) + \lambda \|f\|_K^2 \}$. According to Lemma 2 in Liu et al. (2021) (can be also seen in Li et al. (2019) and Rudi & Rosasco (2017)), one has, when $m = \Omega \left( \lambda^{-2r} \vee \lambda^{-1} \log \frac{1}{\lambda \delta} \right)$, with probability at least $1 - \delta$,

$$\left\| S_m \left(S_m^* S_m + \frac{\lambda}{2} I\right)^{-1} S_m^* f_\rho - \tilde{f}_\lambda \right\|_K \leq c\lambda^r.$$

Combining the above, we complete this proof. $\qquad \square$

### B.2 PROOF OF THEOREM 1

*Proof.* We have

$$
\begin{aligned}
&\left\| \bar{f}_{m,D,\lambda}^0 - f_\rho \right\|_K \\
&= \left\| \bar{f}_{m,D,\lambda}^0 - f_{m,D,\lambda} + f_{m,D,\lambda} - f_{m,\lambda} + f_{m,\lambda} - f_\lambda + f_\lambda - f_\rho \right\|_K \\
&\leq \left\| \bar{f}_{m,D,\lambda}^0 - f_{m,D,\lambda} \right\|_K + \left\| f_{m,D,\lambda} - f_{m,\lambda} \right\|_K + \left\| f_{m,\lambda} - f_\lambda \right\|_K + \left\| f_\lambda - f \right\|_K.
\end{aligned}
\tag{26}
$$

Combining Lemma 1, Lemma 2, and Lemma 7, we have

$$\left\|\bar{f}_{m,D,\lambda}^0 - f_{m,D,\lambda}\right\|_K$$

$$\leq \sum_{j=1}^{p} \frac{|D_j|^2}{\sum_{k=1}^{p} |D_k|^2} \left\|\left(\boldsymbol{\Phi}_{m,D}\mathbf{W}_D\boldsymbol{\Phi}_{m,D}^T + \frac{\lambda}{2}I\right)^{-1/2} SS_\lambda^{1/2}\right\|^2$$

$$* \left(\left\|SS_\lambda^{-1/2}\left(S_m^*S_m - \boldsymbol{\Phi}_{m,D}\mathbf{W}_D\boldsymbol{\Phi}_{m,D}^T\right)\left(S_m^*S_m + \frac{\lambda}{2}I\right)^{-1/2}\right\|\right.$$

$$+ \left.\left\|SS_\lambda^{-1/2}\left(S_m^*S_m - \boldsymbol{\Phi}_{m,D_j}\mathbf{W}_{D_j}\boldsymbol{\Phi}_{m,D_j}^T\right)\left(S_m^*S_m + \frac{\lambda}{2}I\right)^{-1/2}\right\|\right)$$

$$* \left(\left(\sqrt{2}\left\|\left(\boldsymbol{\Phi}_{m,D_j}\mathbf{W}_{D_j}\boldsymbol{\Phi}_{m,D_j}^T + \frac{\lambda}{2}I\right)^{-1/2} SS_\lambda^{1/2}\right\| + \left\|\left(\boldsymbol{\Phi}_{m,D_j}\mathbf{W}_{D_j}\boldsymbol{\Phi}_{m,D_j}^T + \frac{\lambda}{2}I\right)^{-1/2} SS_\lambda^{1/2}\right\|^2\right)$$

$$* \left(\left\|SS_\lambda^{-1/2}\left(\boldsymbol{\Phi}_{m,D_j}\mathbf{W}_{D_j}\bar{\mathbf{y}}_{D_j} - S_m^*\mathbf{W}_{D_j}f_\rho\right)\right\| + \left\|\left(S_m^*S_m + \frac{\lambda}{2}I\right)^{-1/2}\left(S_m^*\mathbf{W}_{D_j}f_\rho - S_{m,D_j}^*\mathbf{W}_{D_j}f_\rho\right)\right\|\right)$$

$$+ \left(2\left\|\left(\boldsymbol{\Phi}_{m,D_j}\mathbf{W}_{D_j}\boldsymbol{\Phi}_{m,D_j}^T + \frac{\lambda}{2}I\right)^{-1/2} SS_\lambda^{1/2}\right\| + \left\|\left(\boldsymbol{\Phi}_{m,D_j}\mathbf{W}_{D_j}\boldsymbol{\Phi}_{m,D_j}^T + \frac{\lambda}{2}I\right)^{-1/2} SS_\lambda^{1/2}\right\|^2 + 1\right)$$

$$* \left\|f_{m,\lambda} - f_\rho\right\|_K\right),$$

where $SS_\lambda = \left(S_m^*S_m + \frac{\lambda}{2}I\right)$.

From Lemma 5, we know that if $|D| \geq 32\log(2/\delta)\left(1 + 2\tau^2\lambda^{-1}\right)$,

$$\left\|\left(S_m^*S_m + \frac{\lambda}{2}I\right)^{-1/2}\left(\boldsymbol{\Phi}_{m,D}\mathbf{W}_D\boldsymbol{\Phi}_{m,D}^T - S_m^*S_m\right)\left(S_m^*S_m + \frac{\lambda}{2}\right)^{-1/2}\right\| \leq \frac{1}{2}.$$

Combining the above inequality and Proposition 2, for any $\delta > 0$, with probability at least $1 - \delta$, we can obtain,

$$\left\|\left(\boldsymbol{\Phi}_{m,D}\mathbf{W}_D\boldsymbol{\Phi}_{m,D}^T + \frac{\lambda}{2}I\right)^{-1/2}\left(S_m^*S_m + \frac{\lambda}{2}I\right)^{1/2}\right\| \leq \sqrt{2}. \tag{27}$$

From Lemma 2, we have

$$\left\|f_{m,D,\lambda} - f_{m,\lambda}\right\|_K$$

$$\leq \left\|\left(\boldsymbol{\Phi}_{m,D}\mathbf{W}_D\boldsymbol{\Phi}_{m,D}^T + \frac{\lambda}{2}I\right)^{-1/2}\left(S_m^*S_m + \frac{\lambda}{2}I\right)^{1/2}\right\|^2$$

$$* \left(\left\|\left(S_m^*S_m + \frac{\lambda}{2}I\right)^{-1/2}\left(\boldsymbol{\Phi}_{m,D}\mathbf{W}_D\bar{\mathbf{y}}_D - S_m^*\mathbf{W}_D f_\rho\right)\right\| + \left\|\left(S_m^*S_m + \frac{\lambda}{2}I\right)^{-1/2}\left(S_m^*\mathbf{W}_D f_\rho - S_{m,D}^*\mathbf{W}_D f_\rho\right)\right\|\right)$$

$$+ \left(\left\|\left(\boldsymbol{\Phi}_{m,D}\mathbf{W}_D\boldsymbol{\Phi}_{m,D}^T + \frac{\lambda}{2}I\right)^{-1/2}\left(S_m^*S_m + \frac{\lambda}{2}I\right)^{1/2}\right\|\right.$$

$$+ \left.\left\|\left(\boldsymbol{\Phi}_{m,D}\mathbf{W}_D\boldsymbol{\Phi}_{m,D}^T + \frac{\lambda}{2}I\right)^{-1/2}\left(S_m^*S_m + \frac{\lambda}{2}I\right)^{1/2}\right\|^2\right)\left\|f_{m,\lambda} - f_\rho\right\|_K.$$

From Proposition 3, Lemma 3, Lemma 4, and Eq.(27), we know that if $|D| \geq \Omega\left(\tau^2\lambda^{-1}\right)$, we have

$$\left\|f_{m,D,\lambda} - f_{m,\lambda}\right\|_K = \mathcal{O}\left(\Upsilon_{m,D,\lambda}\log\frac{1}{\delta} + \left\|f_{m,\lambda} - f_\lambda\right\|_K + \left\|f_\lambda - f_\rho\right\|_K\right), \tag{28}$$

where

$$\Upsilon_{m,D,\lambda} = \mathcal{O}\left(\frac{1}{\sqrt{\lambda|D|}}\right). \tag{29}$$

Note that

$$\left\|\left(S_m^* S_m + \frac{\lambda}{2}I\right)^{-1/2}\left(S_m^* S_m - \mathbf{\Phi}_{m,D}\mathbf{W}_D \mathbf{\Phi}_{m,D}^T\right)\left(S_m^* S_m + \frac{\lambda}{2}I\right)^{-1/2}\right\|$$

$$\leq \left\|\left(S_m^* S_m + \frac{\lambda}{2}I\right)^{-1/2}\left(S_m^* S_m - \mathbf{\Phi}_{m,D_j}\mathbf{W}_{D_j} \mathbf{\Phi}_{m,D_j}^T\right)\left(S_m^* S_m + \frac{\lambda}{2}I\right)^{-1/2}\right\|.$$

According to Proposition 4 and Lemma 8, we have

$$\left\|\bar{f}_{m,D,\lambda}^0 - f_{m,D,\lambda}\right\|_K$$

$$=\mathcal{O}\left(\sum_{j=1}^p \frac{|D_j|^2}{\sum_{k=1}^p |D_k|^2}\left\|\left(S_m^* S_m + \frac{\lambda}{2}I\right)^{-1/2}\left(S_m^* S_m - \mathbf{\Phi}_{m,D_j}\mathbf{W}_{D_j} \mathbf{\Phi}_{m,D_j}^T\right)\left(S_m^* S_m + \frac{\lambda}{2}I\right)^{-1/2}\right\| \Upsilon_{m,D_j,\lambda} \log\frac{1}{\delta}\right.$$

$$\left.+\lambda^r\left\|\left(S_m^* S_m + \frac{\lambda}{2}I\right)^{-1/2}\left(S_m^* S_m - \mathbf{\Phi}_{m,D_j}\mathbf{W}_{D_j} \mathbf{\Phi}_{m,D_j}^T\right)\left(S_m^* S_m + \frac{\lambda}{2}I\right)^{-1/2}\right\|\right). \tag{30}$$

Combining Eq.(26), Eq.(28), Eq.(30), Proposition 4, and Lemma 8, one can obtain, if $m = \Omega\left(\lambda^{-2r} \vee \lambda^{-1}\log\frac{1}{\lambda\delta}\right)$, with probability $1 - \delta$, we have

$$\left\|\bar{f}_{m,D,\lambda}^0 - f_\rho\right\|_K$$

$$=\mathcal{O}\left(\sum_{j=1}^p \frac{|D_j|^2}{\sum_{k=1}^p |D_k|^2}\left\|\left(S_m^* S_m + \frac{\lambda}{2}I\right)^{-1/2}\left(S_m^* S_m - \mathbf{\Phi}_{m,D_j}\mathbf{W}_{D_j} \mathbf{\Phi}_{m,D_j}^T\right)\left(S_m^* S_m + \frac{\lambda}{2}I\right)^{-1/2}\right\| \Upsilon_{m,D_j,\lambda} \log\frac{1}{\delta}\right.$$

$$\left.+\Upsilon_{m,D,\lambda} \log\frac{1}{\delta} + \lambda^r\left\|\left(S_m^* S_m + \frac{\lambda}{2}I\right)^{-1/2}\left(S_m^* S_m - \mathbf{\Phi}_{m,D_j}\mathbf{W}_{D_j} \mathbf{\Phi}_{m,D_j}^T\right)\left(S_m^* S_m + \frac{\lambda}{2}I\right)^{-1/2}\right\| + \lambda^r\right). \tag{31}$$

According to Lemma 5, we have

$$\left\|\left(S_m^* S_m + \frac{\lambda}{2}I\right)^{-1/2}\left(S_m^* S_m - \mathbf{\Phi}_{m,D}\mathbf{W}_D \mathbf{\Phi}_{m,D}^T\right)\left(S_m^* S_m + \frac{\lambda}{2}I\right)^{-1/2}\right\|$$

$$\leq \frac{2\log^2(2/\delta)\left(2\tau^2\lambda^{-1} + 1\right)}{|D|} + \sqrt{\frac{2\log(2/\delta)\left(2\tau^2\lambda^{-1} + 1\right)}{|D|}}. \tag{32}$$

Set $\lambda = \mathcal{O}\left(\left(\sum_{j=1}^p \frac{|D_j|}{\sum_{k=1}^p |D_k|^2}\right)^{\frac{1}{1+r}}\right)$, we have the number of random features

$$m = \Omega\left(\left(\sum_{j=1}^p \frac{|D_j|}{\sum_{k=1}^p |D_k|^2}\right)^{\frac{-2r}{1+r}}\right).$$

Combining Eq.(31), Eq.(29), and Eq.(32), we have

$$\left\|\bar{f}_{m,D,\lambda}^0 - f_\rho\right\|_K = \mathcal{O}\left(\left(\sum_{j=1}^p \frac{|D_j|}{\sum_{k=1}^p |D_k|^2}\right)^{\frac{r}{1+r}} \log\frac{1}{\delta}\right).$$

We complete this proof.

$\square$

## C PROOF OF THEOREM 2

### C.1 BOUND TERMS

**Lemma 9.** *We have*

$$\left\|\bar{f}^l_{m,D,\lambda} - f_{m,D,\lambda}\right\|_K \le \left(\sum_{j=1}^p \frac{|D_j|^2}{\sum_{k=1}^p |D_k|^2} \mathcal{J}_m\right)^l \left(\left\|\bar{f}^0_{m,D,\lambda} - f_{m,D,\lambda}\right\|_K + \sqrt{\lambda}\left\|\bar{g}^0_{m,D,\lambda} - g_{m,D,\lambda}\right\|\right),$$

*where*

$$\begin{aligned}
\mathcal{J}_m =& 2\left\|\left(\mathbf{\Phi}_{m,D_j}\mathbf{W}_{D_j}\mathbf{\Phi}^T_{m,D_j} + \frac{\lambda}{2}I\right)^{-1/2}\left(S_m^* S_m + \frac{\lambda}{2}I\right)^{1/2}\right\|^2 \\
&* \left\|\left(S_m^* S_m + \frac{\lambda}{2}I\right)^{-1/2}\left(S_m^* S_m - \mathbf{\Phi}_{m,D}\mathbf{W}_D\mathbf{\Phi}^T_{m,D}\right)\left(S_m^* S_m + \frac{\lambda}{2}I\right)^{-1/2}\right\| \\
&+ 2\left\|\left(\mathbf{\Phi}_{m,D_j}\mathbf{W}_{D_j}\mathbf{\Phi}^T_{m,D_j} + \frac{\lambda}{2}I\right)^{-1/2}\left(S_m^* S_m + \frac{\lambda}{2}I\right)^{1/2}\right\|^2 \\
&* \left\|\left(S_m^* S_m + \frac{\lambda}{2}I\right)^{-1/2}\left(S_m^* S_m - \mathbf{\Phi}_{m,D_j}\mathbf{W}_{D_j}\mathbf{\Phi}^T_{m,D_j}\right)\left(S_m^* S_m + \frac{\lambda}{2}I\right)^{-1/2}\right\|.
\end{aligned}$$

*Proof.* Note that

$$\mathbf{g}_{m,D,\lambda} = \bar{\mathbf{g}}^{l-1}_{m,D,\lambda} - \left(\mathbf{\Phi}_{m,D}\mathbf{W}_D\mathbf{\Phi}^T_{m,D} + \frac{\lambda}{2}I\right)^{-1}\left[\left(\mathbf{\Phi}_{m,D}\mathbf{W}_D\mathbf{\Phi}^T_{m,D} + \frac{\lambda}{2}I\right)\bar{\mathbf{g}}^{l-1}_{m,D,\lambda} - \mathbf{\Phi}_{m,D}\mathbf{W}_D\bar{\mathbf{y}}_D\right],$$

and

$$\begin{aligned}
&\bar{\mathbf{g}}^l_{m,D,\lambda} \\
=& \bar{\mathbf{g}}^{l-1}_{m,D,\lambda} - \sum_{j=1}^p \frac{|D_j|^2}{\sum_{k=1}^p |D_k|^2}\left(\mathbf{\Phi}_{m,D_j}\mathbf{W}_{D_j}\mathbf{\Phi}^T_{m,D_j} + \frac{\lambda}{2}I\right)^{-1}\left[\left(\mathbf{\Phi}_{m,D}\mathbf{W}_D\mathbf{\Phi}^T_{m,D} + \frac{\lambda}{2}I\right)\bar{\mathbf{g}}^{l-1}_{m,D,\lambda} - \mathbf{\Phi}_{m,D}\mathbf{W}_D\bar{\mathbf{y}}_D\right].
\end{aligned}$$

Thus, we have

$$\begin{aligned}
&\mathbf{g}_{m,D,\lambda} - \bar{\mathbf{g}}^l_{m,D,\lambda} \\
=& \bar{\mathbf{g}}^{l-1}_{m,D,\lambda} - \left(\mathbf{\Phi}_{m,D}\mathbf{W}_D\mathbf{\Phi}^T_{m,D} + \frac{\lambda}{2}I\right)^{-1}\left[\left(\mathbf{\Phi}_{m,D}\mathbf{W}_D\mathbf{\Phi}^T_{m,D} + \frac{\lambda}{2}I\right)\bar{\mathbf{g}}^{l-1}_{m,D,\lambda} - \mathbf{\Phi}_{m,D}\mathbf{W}_D\bar{\mathbf{y}}_D\right] \\
&- \bar{\mathbf{g}}^{l-1}_{m,D,\lambda} + \sum_{j=1}^p \frac{|D_j|^2}{\sum_{k=1}^p |D_k|^2}\left(\mathbf{\Phi}_{m,D_j}\mathbf{W}_{D_j}\mathbf{\Phi}^T_{m,D_j} + \frac{\lambda}{2}I\right)^{-1} \\
&* \left[\left(\mathbf{\Phi}_{m,D}\mathbf{W}_D\mathbf{\Phi}^T_{m,D} + \frac{\lambda}{2}I\right)\bar{\mathbf{g}}^{l-1}_{m,D,\lambda} - \mathbf{\Phi}_{m,D}\mathbf{W}_D\bar{\mathbf{y}}_D\right] \\
=& \sum_{j=1}^p \frac{|D_j|^2}{\sum_{k=1}^p |D_k|^2}\left[\left(\mathbf{\Phi}_{m,D_j}\mathbf{W}_{D_j}\mathbf{\Phi}^T_{m,D_j} + \frac{\lambda}{2}I\right)^{-1} - \left(\mathbf{\Phi}_{m,D}\mathbf{W}_D\mathbf{\Phi}^T_{m,D} + \frac{\lambda}{2}I\right)^{-1}\right] \\
&* \left[\left(\mathbf{\Phi}_{m,D}\mathbf{W}_D\mathbf{\Phi}^T_{m,D} + \frac{\lambda}{2}I\right)\bar{\mathbf{g}}^{l-1}_{m,D,\lambda} - \mathbf{\Phi}_{m,D}\mathbf{W}_D\bar{\mathbf{y}}_D\right].
\end{aligned}$$

$$(33)$$

The above can be convert into

$$
\begin{aligned}
&\mathbf{g}_{m,D,\lambda} - \bar{\mathbf{g}}_{m,D,\lambda}^{l} \\
&= \sum_{j=1}^{p} \frac{|D_j|^2}{\sum_{k=1}^{p}|D_k|^2} \left( \mathbf{\Phi}_{m,D_j} \mathbf{W}_{D_j} \mathbf{\Phi}_{m,D_j}^{T} + \frac{\lambda}{2}I \right)^{-1} \left[ \mathbf{\Phi}_{m,D}\mathbf{W}_D\mathbf{\Phi}_{m,D}^{T} - \mathbf{\Phi}_{m,D_j}\mathbf{W}_{D_j}\mathbf{\Phi}_{m,D_j}^{T} \right] \\
&\quad * \left( \mathbf{\Phi}_{m,D}\mathbf{W}_D\mathbf{\Phi}_{m,D}^{T} + \frac{\lambda}{2}I \right)^{-1} \left[ \left( \mathbf{\Phi}_{m,D}\mathbf{W}_D\mathbf{\Phi}_{m,D}^{T} + \frac{\lambda}{2}I \right) \bar{\mathbf{g}}_{m,D,\lambda}^{l-1} - \mathbf{\Phi}_{m,D}\mathbf{W}_D\bar{\mathbf{y}}_D \right] \\
&= \sum_{j=1}^{p} \frac{|D_j|^2}{\sum_{k=1}^{p}|D_k|^2} \left( \mathbf{\Phi}_{m,D_j} \mathbf{W}_{D_j} \mathbf{\Phi}_{m,D_j}^{T} + \frac{\lambda}{2}I \right)^{-1} \left[ \mathbf{\Phi}_{m,D}\mathbf{W}_D\mathbf{\Phi}_{m,D}^{T} - \mathbf{\Phi}_{m,D_j}\mathbf{W}_{D_j}\mathbf{\Phi}_{m,D_j}^{T} \right] \\
&\quad * \left[ \bar{\mathbf{g}}_{m,D,\lambda}^{l-1} - \mathbf{g}_{m,D,\lambda} \right] \\
&= \underbrace{\sum_{j=1}^{p} \frac{|D_j|^2}{\sum_{k=1}^{p}|D_k|^2} \left( \mathbf{\Phi}_{m,D_j} \mathbf{W}_{D_j} \mathbf{\Phi}_{m,D_j}^{T} + \frac{\lambda}{2}I \right)^{-1} \left[ \mathbf{\Phi}_{m,D}\mathbf{W}_D\mathbf{\Phi}_{m,D}^{T} - S_m^* S_m \right] \left[ \bar{\mathbf{g}}_{m,D,\lambda}^{l-1} - \mathbf{g}_{m,D,\lambda} \right]}_{\text{Term-A}} \\
&\quad + \underbrace{\sum_{j=1}^{p} \frac{|D_j|^2}{\sum_{k=1}^{p}|D_k|^2} \left( \mathbf{\Phi}_{m,D_j} \mathbf{W}_{D_j} \mathbf{\Phi}_{m,D_j}^{T} + \frac{\lambda}{2}I \right)^{-1} \left[ S_m^* S_m - \mathbf{\Phi}_{m,D_j}\mathbf{W}_{D_j}\mathbf{\Phi}_{m,D_j}^{T} \right] \left[ \bar{\mathbf{g}}_{m,D,\lambda}^{l-1} - \mathbf{g}_{m,D,\lambda} \right]}_{\text{Term-B}} .
\end{aligned}
\tag{34}
$$

Note that

$$
\begin{aligned}
&S_m * \textbf{Term-A} \\
&= S_m \left( S_m^* S_m + \frac{\lambda}{2}I \right)^{-1/2} \left( S_m^* S_m + \frac{\lambda}{2}I \right)^{1/2} \left( \mathbf{\Phi}_{m,D_j} \mathbf{W}_{D_j} \mathbf{\Phi}_{m,D_j}^{T} + \frac{\lambda}{2}I \right)^{-1/2} \\
&\quad * \left( \mathbf{\Phi}_{m,D_j} \mathbf{W}_{D_j} \mathbf{\Phi}_{m,D_j}^{T} + \frac{\lambda}{2}I \right)^{-1/2} \left( S_m^* S_m + \frac{\lambda}{2}I \right)^{1/2} \left( S_m^* S_m + \frac{\lambda}{2}I \right)^{-1/2} \\
&\quad * \left[ \mathbf{\Phi}_{m,D}\mathbf{W}_D\mathbf{\Phi}_{m,D}^{T} - S_m^* S_m \right] \left( S_m^* S_m + \frac{\lambda}{2}I \right)^{-1/2} \\
&\quad * \left( S_m^* S_m + \frac{\lambda}{2}I \right)^{-1/2} \left( S_m^* S_m + \frac{\lambda}{2}I \right) \left( \bar{\mathbf{g}}_{m,D,\lambda}^{l-1} - \mathbf{g}_{m,D,\lambda} \right) .
\end{aligned}
$$

Note that $\left\| S_m \left( S_m^* S_m + \frac{\lambda}{2}I \right)^{-1/2} \right\| = \left\| \left( S_m^* S_m + \frac{\lambda}{2}I \right)^{-1/2} S_m^* S_m \left( S_m^* S_m + \frac{\lambda}{2}I \right)^{-1/2} \right\|^{1/2} \leq 1$, so, we have

$$
\begin{aligned}
&\| S_m * \textbf{Term-A} \|_K \\
&\leq \left\| \left( \mathbf{\Phi}_{m,D_j} \mathbf{W}_{D_j} \mathbf{\Phi}_{m,D_j}^{T} + \frac{\lambda}{2}I \right)^{-1/2} \left( S_m^* S_m + \frac{\lambda}{2}I \right)^{1/2} \right\|^2 \\
&\quad * \left\| \left( S_m^* S_m + \frac{\lambda}{2}I \right)^{-1/2} \left( S_m^* S_m - \mathbf{\Phi}_{m,D}\mathbf{W}_D\mathbf{\Phi}_{m,D}^{T} \right) \left( S_m^* S_m + \frac{\lambda}{2}I \right)^{-1/2} \right\| \\
&\quad * \left\| \left( S_m^* S_m + \frac{\lambda}{2}I \right)^{-1/2} \left( S_m^* S_m + \frac{\lambda}{2}I \right) \left( \bar{\mathbf{g}}_{m,D,\lambda}^{l-1} - \mathbf{g}_{m,D,\lambda} \right) \right\| .
\end{aligned}
\tag{35}
$$

Note that

$$
S_m^* S_m \left( \bar{\mathbf{g}}_{m,D,\lambda}^{l-1} - \mathbf{g}_{m,D,\lambda} \right) = S_m^* \left( \bar{f}_{m,D,\lambda}^{l-1} - f_{m,D,\lambda} \right) .
$$

Substituting the above into Eq.(35), we have

$$
\begin{aligned}
&\|S_m * \textbf{Term-A}\|_K \\
&\leq \left\|\left(\boldsymbol{\Phi}_{m,D_j}\mathbf{W}_{D_j}\boldsymbol{\Phi}_{m,D_j}^T + \frac{\lambda}{2}I\right)^{-1/2}\left(S_m^*S_m + \frac{\lambda}{2}I\right)^{1/2}\right\|^2 \\
&\quad * \left\|\left(S_m^*S_m + \frac{\lambda}{2}I\right)^{-1/2}(S_m^*S_m - \boldsymbol{\Phi}_{m,D}\mathbf{W}_D\boldsymbol{\Phi}_{m,D}^T)\left(S_m^*S_m + \frac{\lambda}{2}I\right)^{-1/2}\right\| \\
&\quad * \left\|\left(S_m^*S_m + \frac{\lambda}{2}I\right)^{-1/2}S_m^*\left(\bar{f}_{m,D,\lambda}^{l-1} - f_{m,D,\lambda}\right)\right\| \\
&\quad + \frac{\lambda}{2}\left\|\left(\boldsymbol{\Phi}_{m,D_j}\mathbf{W}_{D_j}\boldsymbol{\Phi}_{m,D_j}^T + \frac{\lambda}{2}I\right)^{-1/2}\left(S_m^*S_m + \frac{\lambda}{2}I\right)^{1/2}\right\|^2 \\
&\quad * \left\|\left(S_m^*S_m + \frac{\lambda}{2}I\right)^{-1/2}(S_m^*S_m - \boldsymbol{\Phi}_{m,D}\mathbf{W}_D\boldsymbol{\Phi}_{m,D}^T)\left(S_m^*S_m + \frac{\lambda}{2}I\right)^{-1/2}\right\| \\
&\quad * \left\|\left(S_m^*S_m + \frac{\lambda}{2}I\right)^{-1/2}\left(\bar{\mathbf{g}}_{m,D,\lambda}^{l-1} - \mathbf{g}_{m,D,\lambda}\right)\right\| \\
&\leq \left\|\left(\boldsymbol{\Phi}_{m,D_j}\mathbf{W}_{D_j}\boldsymbol{\Phi}_{m,D_j}^T + \frac{\lambda}{2}I\right)^{-1/2}\left(S_m^*S_m + \frac{\lambda}{2}I\right)^{1/2}\right\|^2 \\
&\quad * \left\|\left(S_m^*S_m + \frac{\lambda}{2}I\right)^{-1/2}(S_m^*S_m - \boldsymbol{\Phi}_{m,D}\mathbf{W}_D\boldsymbol{\Phi}_{m,D}^T)\left(S_m^*S_m + \frac{\lambda}{2}I\right)^{-1/2}\right\| \\
&\quad * \left(\left\|\bar{f}_{m,D,\lambda}^{l-1} - f_{m,D,\lambda}\right\|_K + \sqrt{\lambda}\left\|\bar{\mathbf{g}}_{m,D,\lambda}^{l-1} - \mathbf{g}_{m,D,\lambda}\right\|\right),
\end{aligned}
$$

the last inequality use the fact that

$$
\left\|\left(S_m^*S_m + \frac{\lambda}{2}I\right)^{-1/2}S_m^*\right\| = \left\|\left(S_m^*S_m + \frac{\lambda}{2}I\right)^{-1/2}S_m^*S_m\left(S_m^*S_m + \lambda I\right)^{-1/2}\right\|^{1/2} \leq 1.
$$

Using the same process, we can obtain that

$$
\begin{aligned}
&\|S_m * \textbf{Term-B}\|_K \\
&\leq \left\|\left(\boldsymbol{\Phi}_{m,D_j}\mathbf{W}_{D_j}\boldsymbol{\Phi}_{m,D_j}^T + \frac{\lambda}{2}I\right)^{-1/2}\left(S_m^*S_m + \frac{\lambda}{2}I\right)^{1/2}\right\|^2 \\
&\quad * \left\|\left(S_m^*S_m + \frac{\lambda}{2}I\right)^{-1/2}\left(S_m^*S_m - \boldsymbol{\Phi}_{m,D_j}\mathbf{W}_{D_j}\boldsymbol{\Phi}_{m,D_j}^T\right)\left(S_m^*S_m + \frac{\lambda}{2}I\right)^{-1/2}\right\| \\
&\quad * \left(\left\|\bar{f}_{m,D,\lambda}^{l-1} - f_{m,D,\lambda}\right\|_K + \sqrt{\lambda}\left\|\bar{\mathbf{g}}_{m,D,\lambda}^{l-1} - \mathbf{g}_{m,D,\lambda}\right\|\right).
\end{aligned}
$$

Thus, we have

$$
\begin{aligned}
&\left\| f_{m,D,\lambda} - \bar{f}^l_{m,D,\lambda} \right\|_K \\
&= \left\| S_m \left( \mathbf{g}_{m,D,\lambda} - \bar{\mathbf{g}}^l_{m,D,\lambda} \right) \right\|_K \\
&\leq \sum_{j=1}^p \frac{|D_j|^2}{\sum_{k=1}^p |D_k|^2} \left\| S_m * \mathbf{Term\text{-}A} \right\|_K + \left\| S_m * \mathbf{Term\text{-}B} \right\|_K \\
&\leq \sum_{j=1}^p \frac{|D_j|^2}{\sum_{k=1}^p |D_k|^2} \left( \left\| \left( \mathbf{\Phi}_{m,D_j} \mathbf{W}_{D_j} \mathbf{\Phi}^T_{m,D_j} + \frac{\lambda}{2} I \right)^{-1/2} \left( S_m^* S_m + \frac{\lambda}{2} I \right)^{1/2} \right\|^2 \right. \\
&\qquad * \left\| \left( S_m^* S_m + \frac{\lambda}{2} I \right)^{-1/2} \left( S_m^* S_m - \mathbf{\Phi}_{m,D} \mathbf{W}_D \mathbf{\Phi}^T_{m,D} \right) \left( S_m^* S_m + \frac{\lambda}{2} I \right)^{-1/2} \right\| \\
&\qquad + \left\| \left( \mathbf{\Phi}_{m,D_j} \mathbf{W}_{D_j} \mathbf{\Phi}^T_{m,D_j} + \frac{\lambda}{2} I \right)^{-1/2} \left( S_m^* S_m + \frac{\lambda}{2} I \right)^{1/2} \right\|^2 \\
&\qquad \left. * \left\| \left( S_m^* S_m + \frac{\lambda}{2} I \right)^{-1/2} \left( S_m^* S_m - \mathbf{\Phi}_{m,D_j} \mathbf{W}_{D_j} \mathbf{\Phi}^T_{m,D_j} \right) \left( S_m^* S_m + \frac{\lambda}{2} I \right)^{-1/2} \right\| \right) \\
&\qquad * \left( \left\| \bar{f}^{l-1}_{m,D,\lambda} - f_{m,D,\lambda} \right\|_K + \sqrt{\lambda} \left\| \bar{\mathbf{g}}^{l-1}_{m,D,\lambda} - \mathbf{g}_{m,D,\lambda} \right\| \right).
\end{aligned}
\tag{36}
$$

According to Eq.(34), we know that

$$
\begin{aligned}
&\mathbf{g}_{m,D,\lambda} - \bar{\mathbf{g}}^l_{m,D,\lambda} \\
&= \sum_{j=1}^p \frac{|D_j|^2}{\sum_{k=1}^p |D_k|^2} \left( \mathbf{Term\text{-}A} + \mathbf{Term\text{-}B} \right) \\
&= \sum_{j=1}^p \frac{|D_j|^2}{\sum_{k=1}^p |D_k|^2} \left( \mathbf{\Phi}_{m,D_j} \mathbf{W}_{D_j} \mathbf{\Phi}^T_{m,D_j} + \frac{\lambda}{2} I \right)^{-1} \left[ \mathbf{\Phi}_{m,D} \mathbf{W}_D \mathbf{\Phi}^T_{m,D} - S_m^* S_m \right] \left[ \bar{\mathbf{g}}^{l-1}_{m,D,\lambda} - \mathbf{g}_{m,D,\lambda} \right] \\
&\quad + \sum_{j=1}^p \frac{|D_j|^2}{\sum_{k=1}^p |D_k|^2} \left( \mathbf{\Phi}_{m,D_j} \mathbf{W}_{D_j} \mathbf{\Phi}^T_{m,D_j} + \frac{\lambda}{2} I \right)^{-1} \left[ S_m^* S_m - \mathbf{\Phi}_{m,D_j} \mathbf{W}_{D_j} \mathbf{\Phi}^T_{m,D_j} \right] \left[ \bar{\mathbf{g}}^{l-1}_{m,D,\lambda} - \mathbf{g}_{m,D,\lambda} \right] \\
&= \sum_{j=1}^p \frac{|D_j|^2}{\sum_{k=1}^p |D_k|^2} \left( \mathbf{\Phi}_{m,D_j} \mathbf{W}_{D_j} \mathbf{\Phi}^T_{m,D_j} + \frac{\lambda}{2} I \right)^{-1} \left( S_m^* S_m + \frac{\lambda}{2} I \right) \left( S_m^* S_m + \frac{\lambda}{2} I \right)^{-1} \\
&\quad * \left[ \mathbf{\Phi}_{m,D} \mathbf{W}_D \mathbf{\Phi}^T_{m,D} - S_m^* S_m \right] \left[ \bar{\mathbf{g}}^{l-1}_{m,D,\lambda} - \mathbf{g}_{m,D,\lambda} \right] \\
&\quad + \sum_{j=1}^p \frac{|D_j|^2}{\sum_{k=1}^p |D_k|^2} \left( \mathbf{\Phi}_{m,D_j} \mathbf{W}_{D_j} \mathbf{\Phi}^T_{m,D_j} + \frac{\lambda}{2} I \right)^{-1} \left( S_m^* S_m + \frac{\lambda}{2} I \right) \left( S_m^* S_m + \frac{\lambda}{2} I \right)^{-1} \\
&\quad * \left[ S_m^* S_m - \mathbf{\Phi}_{m,D_j} \mathbf{W}_{D_j} \mathbf{\Phi}^T_{m,D_j} \right] \left[ \bar{\mathbf{g}}^{l-1}_{m,D,\lambda} - \mathbf{g}_{m,D,\lambda} \right].
\end{aligned}
$$

Thus, we obtain that

$$
\begin{aligned}
&\left\| \mathbf{g}_{m,D,\lambda} - \bar{\mathbf{g}}^l_{m,D,\lambda} \right\| \\
&\leq \sum_{j=1}^p \frac{|D_j|^2}{\sum_{k=1}^p |D_k|^2} \left\| \left( \mathbf{\Phi}_{m,D_j} \mathbf{W}_{D_j} \mathbf{\Phi}^T_{m,D_j} + \frac{\lambda}{2} I \right)^{-1/2} \left( S_m^* S_m + \frac{\lambda}{2} I \right)^{1/2} \right\|^2 \\
&\quad * \left( \left\| \left( S_m^* S_m + \frac{\lambda}{2} I \right)^{-1/2} \left( S_m^* S_m - \mathbf{\Phi}_{m,D} \mathbf{W}_D \mathbf{\Phi}^T_{m,D} \right) \left( S_m^* S_m + \frac{\lambda}{2} I \right)^{-1/2} \right\| \right. \\
&\quad + \left. \left\| \left( S_m^* S_m + \frac{\lambda}{2} I \right)^{-1/2} \left( S_m^* S_m - \mathbf{\Phi}_{m,D_j} \mathbf{W}_{D_j} \mathbf{\Phi}^T_{m,D_j} \right) \left( S_m^* S_m + \frac{\lambda}{2} I \right)^{-1/2} \right\| \right) \\
&\quad * \left\| \bar{\mathbf{g}}^{l-1}_{m,D,\lambda} - \mathbf{g}_{m,D,\lambda} \right\|.
\end{aligned}
\tag{37}
$$

Combining Eq.(36) and Eq.(37), we have

$$\left\| f_{m,D,\lambda} - \bar{f}^l_{m,D,\lambda} \right\|_K + \sqrt{\lambda} \left\| \mathbf{g}_{m,D,\lambda} - \bar{\mathbf{g}}^l_{m,D,\lambda} \right\|$$

$$\leq \sum_{j=1}^p \frac{|D_j|^2}{\sum_{k=1}^p |D_k|^2} \left( \left\| \left( \boldsymbol{\Phi}_{m,D_j} \mathbf{W}_{D_j} \boldsymbol{\Phi}^T_{m,D_j} + \frac{\lambda}{2} I \right)^{-1/2} \left( S_m^* S_m + \frac{\lambda}{2} I \right)^{1/2} \right\|^2 \right.$$

$$* \left\| \left( S_m^* S_m + \frac{\lambda}{2} I \right)^{-1/2} \left( S_m^* S_m - \boldsymbol{\Phi}_{m,D} \mathbf{W}_D \boldsymbol{\Phi}^T_{m,D} \right) \left( S_m^* S_m + \frac{\lambda}{2} I \right)^{-1/2} \right\|$$

$$+ \left\| \left( \boldsymbol{\Phi}_{m,D_j} \mathbf{W}_{D_j} \boldsymbol{\Phi}^T_{m,D_j} + \frac{\lambda}{2} I \right)^{-1/2} \left( S_m^* S_m + \frac{\lambda}{2} I \right)^{1/2} \right\|^2$$

$$\left. * \left\| \left( S_m^* S_m + \frac{\lambda}{2} I \right)^{-1/2} \left( S_m^* S_m - \boldsymbol{\Phi}_{m,D_j} \mathbf{W}_{D_j} \boldsymbol{\Phi}^T_{m,D_j} \right) \left( S_m^* S_m + \frac{\lambda}{2} I \right)^{-1/2} \right\| \right)$$

$$* \left( \left\| \bar{f}^{l-1}_{m,D,\lambda} - f_{m,D,\lambda} \right\|_K + \sqrt{\lambda} \left\| \bar{\mathbf{g}}^{l-1}_{m,D,\lambda} - \mathbf{g}_{m,D,\lambda} \right\| \right)$$

$$+ \sum_{j=1}^p \frac{|D_j|^2}{\sum_{k=1}^p |D_k|^2} \left\| \left( \boldsymbol{\Phi}_{m,D_j} \mathbf{W}_{D_j} \boldsymbol{\Phi}^T_{m,D_j} + \frac{\lambda}{2} I \right)^{-1/2} \left( S_m^* S_m + \frac{\lambda}{2} I \right)^{1/2} \right\|^2$$

$$* \left( \left\| \left( S_m^* S_m + \frac{\lambda}{2} I \right)^{-1/2} \left( S_m^* S_m - \boldsymbol{\Phi}_{m,D} \mathbf{W}_D \boldsymbol{\Phi}^T_{m,D} \right) \left( S_m^* S_m + \frac{\lambda}{2} I \right)^{-1/2} \right\| \right.$$

$$\left. + \left\| \left( S_m^* S_m + \frac{\lambda}{2} I \right)^{-1/2} \left( S_m^* S_m - \boldsymbol{\Phi}_{m,D_j} \mathbf{W}_{D_j} \boldsymbol{\Phi}^T_{m,D_j} \right) \left( S_m^* S_m + \frac{\lambda}{2} I \right)^{-1/2} \right\| \right) \sqrt{\lambda} \left\| \bar{\mathbf{g}}^{l-1}_{m,D,\lambda} - \mathbf{g}_{m,D,\lambda} \right\|$$

$$\leq \sum_{j=1}^p \frac{|D_j|^2}{\sum_{k=1}^p |D_k|^2} \left( 2 \left\| \left( \boldsymbol{\Phi}_{m,D_j} \mathbf{W}_{D_j} \boldsymbol{\Phi}^T_{m,D_j} + \frac{\lambda}{2} I \right)^{-1/2} \left( S_m^* S_m + \frac{\lambda}{2} I \right)^{1/2} \right\|^2 \right.$$

$$* \left\| \left( S_m^* S_m + \frac{\lambda}{2} I \right)^{-1/2} \left( S_m^* S_m - \boldsymbol{\Phi}_{m,D_j} \mathbf{W}_{D_j} \boldsymbol{\Phi}^T_{m,D_j} \right) \left( S_m^* S_m + \frac{\lambda}{2} I \right)^{-1/2} \right\|$$

$$+ 2 \left\| \left( \boldsymbol{\Phi}_{m,D_j} \mathbf{W}_{D_j} \boldsymbol{\Phi}^T_{m,D_j} + \frac{\lambda}{2} I \right)^{-1/2} \left( S_m^* S_m + \frac{\lambda}{2} I \right)^{1/2} \right\|^2$$

$$\left. * \left\| \left( S_m^* S_m + \frac{\lambda}{2} I \right)^{-1/2} \left( S_m^* S_m - \boldsymbol{\Phi}_{m,D_j} \mathbf{W}_{D_j} \boldsymbol{\Phi}^T_{m,D_j} \right) \left( S_m^* S_m + \frac{\lambda}{2} I \right)^{-1/2} \right\| \right)$$

$$* \left( \left\| \bar{f}^{l-1}_{m,D,\lambda} - f_{m,D,\lambda} \right\|_K + \sqrt{\lambda} \left\| \bar{\mathbf{g}}^{l-1}_{m,D,\lambda} - \mathbf{g}_{m,D,\lambda} \right\| \right)$$

$$\leq \left( 2 \sum_{j=1}^p \frac{|D_j|^2}{\sum_{k=1}^p |D_k|^2} \left\| \left( \boldsymbol{\Phi}_{m,D_j} \mathbf{W}_{D_j} \boldsymbol{\Phi}^T_{m,D_j} + \frac{\lambda}{2} I \right)^{-1/2} \left( S_m^* S_m + \frac{\lambda}{2} I \right)^{1/2} \right\|^2 \right.$$

$$* \left\| \left( S_m^* S_m + \frac{\lambda}{2} I \right)^{-1/2} \left( S_m^* S_m - \boldsymbol{\Phi}_{m,D} \mathbf{W}_D \boldsymbol{\Phi}^T_{m,D} \right) \left( S_m^* S_m + \frac{\lambda}{2} I \right)^{-1/2} \right\|$$

$$+ \left\| \left( \boldsymbol{\Phi}_{m,D_j} \mathbf{W}_{D_j} \boldsymbol{\Phi}^T_{m,D_j} + \frac{\lambda}{2} I \right)^{-1/2} \left( S_m^* S_m + \frac{\lambda}{2} I \right)^{1/2} \right\|^2$$

$$\left. * \left\| S S_\lambda^{-1/2} \left( S_m^* S_m - \boldsymbol{\Phi}_{m,D_j} \mathbf{W}_{D_j} \boldsymbol{\Phi}^T_{m,D_j} \right) \left( S_m^* S_m + \frac{\lambda}{2} I \right)^{-1/2} \right\| \right)^l$$

$$* \left( \left\| \bar{f}^0_{m,D,\lambda} - f_{m,D,\lambda} \right\|_K + \sqrt{\lambda} \left\| \bar{\mathbf{g}}^0_{m,D,\lambda} - \mathbf{g}_{m,D,\lambda} \right\| \right).$$

$$\square$$

## C.2 PROOF OF THEOREM 2

*Proof.* Note that

$$\left\|\bar{f}^l_{m,D,\lambda} - f_\rho\right\|_K = \left\|\bar{f}^l_{m,D,\lambda} - f_{m,D,\lambda} + f_{m,D,\lambda} - f_{m,\lambda} + f_{m,\lambda} - f_\lambda + f_\lambda - f_\rho\right\|_K$$

$$\leq \left\|\bar{f}^l_{m,D,\lambda} - f_{m,D,\lambda}\right\|_K + \left\|f_{m,D,\lambda} - f_{m,\lambda}\right\|_K + \left\|f_{m,\lambda} - f_\lambda\right\|_K + \left\|f_\lambda - f\right\|_K. \tag{38}$$

Substituting Lemma 1, Lemma 2, Lemma 3, Lemma 4, Eq.(27), and Eq.(28) into Lemma 6 and Lemma 7, we have

$$\left\|\bar{f}^0_{m,D,\lambda} - f_{m,D,\lambda}\right\|_K + \sqrt{\lambda}\left\|\bar{\mathbf{g}}^0_{D,\lambda} - \mathbf{g}_{m,D,\lambda}\right\|_2$$

$$=\mathcal{O}\left(\sum_{j=1}^p \frac{|D_j|^2}{\sum_{k=1}^p |D_k|^2}\left(\mathcal{K}_{m,D} + \mathcal{K}_{m,D_j}\right)\right.$$

$$\left.* \left(\left\|SS_\lambda^{-1/2}\left(\mathbf{\Phi}_{m,D}\mathbf{W}_D\bar{\mathbf{y}}_D - S_m^*\mathbf{W}_D f_\rho\right)\right\| + \left\|SS_\lambda^{-1/2}(S_m^*\mathbf{W}_D f_\rho - S_{m,D}^*\mathbf{W}_D f_\rho)\right\| + \|f_{m,\lambda} - f_\rho\|_K\right)\right)$$

$$=\mathcal{O}\left(\sum_{j=1}^p \frac{|D_j|^2}{\sum_{k=1}^p |D_k|^2}\left(\mathcal{K}_{m,D_j} + \mathcal{K}_{m,D_j}\right)\mathcal{Q}_m\right),$$

where $SS_\lambda = \left(S_m^*S_m + \frac{\lambda}{2}I\right), \mathcal{K}_{m,D} = \left\|\left(S_m^*S_m + \frac{\lambda}{2}I\right)^{-1/2}\left(S_m^*S_m - \mathbf{\Phi}_{m,D}\mathbf{W}_D\mathbf{\Phi}_{m,D}^T\right)\left(S_m^*S_m + \frac{\lambda}{2}I\right)^{-1/2}\right\|,$ and $\mathcal{Q}_m = \left(\Upsilon_{m,D_j,\lambda} + \|f_{m,\lambda} - f_\lambda\| + \|f_\lambda - f_\rho\|_K\right).$

Combining the above inequality and Lemma 9, and note that

$$\left\|SS_\lambda^{-1/2}\left(S_m^*S_m - \mathbf{\Phi}_{m,D}\mathbf{W}_D\mathbf{\Phi}_{m,D}^T\right)\left(S_m^*S_m + \frac{\lambda}{2}I\right)^{-1/2}\right\| \leq \left\|SS_\lambda^{-1/2}\left(S_m^*S_m - \mathbf{\Phi}_{m,D_j}\mathbf{W}_{D_j}\mathbf{\Phi}_{m,D_j}^T\right)SS_\lambda^{-1/2}\right\|,$$

we can obtain that

$$\left\|\bar{f}^l_{m,D,\lambda} - f_{m,D,\lambda}\right\|_K$$

$$=\mathcal{O}\left(\left(\sum_{j=1}^p \frac{|D_j|^2}{\sum_{k=1}^p |D_k|^2}\left\|SS_\lambda^{-1/2}\left(S_m^*S_m - \mathbf{\Phi}_{m,D_j}\mathbf{W}_{D_j}\mathbf{\Phi}_{m,D_j}^T\right)SS_\lambda^{-1/2}\right\|\right)^l\right.$$

$$\left.* \left(\sum_{j=1}^p \frac{|D_j|^2}{\sum_{k=1}^p |D_k|^2}\left\|SS_\lambda^{-1/2}\left(S_m^*S_m - \mathbf{\Phi}_{m,D_j}\mathbf{W}_{D_j}\mathbf{\Phi}_{m,D_j}^T\right)SS_\lambda^{-1/2}\right\|\mathcal{Q}_m\right)\right). \tag{39}$$

Combining Eq.(38), Eq.(39), Proposition 4, and Lemma 8, one can obtain, if $m = \Omega\left(\lambda^{-2r} \vee \lambda^{-1}\log\frac{1}{\lambda\delta}\right)$, with probability $1 - \delta$, we have

$$\left\|\bar{f}^l_{m,D,\lambda} - f_\rho\right\|_K$$

$$=\mathcal{O}\left(\left(\sum_{j=1}^p \frac{|D_j|^2}{\sum_{k=1}^p |D_k|^2}\left\|SS_\lambda^{-1/2}\left(S_m^*S_m - \mathbf{\Phi}_{m,D_j}\mathbf{W}_{D_j}\mathbf{\Phi}_{m,D_j}^T\right)SS_\lambda^{-1/2}\right\|\right)^l\right.$$

$$* \left(\sum_{j=1}^p \frac{|D_j|^2}{\sum_{k=1}^p |D_k|^2}\left\|SS_\lambda^{-1/2}\left(S_m^*S_m - \mathbf{\Phi}_{m,D_j}\mathbf{W}_{D_j}\mathbf{\Phi}_{m,D_j}^T\right)SS_\lambda^{-1/2}\right\|\left(\Upsilon_{m,D_j,\lambda} + \lambda^r\right)\right)$$

$$\left. + \Upsilon_{m,D,\lambda}\log\frac{1}{\delta} + \lambda^r\right).$$

Set $\lambda = \mathcal{O}(|D|^{-\frac{1}{1+r}})$, $|D_1| = \ldots = |D_p| = \frac{|D|}{p}$, and the number of random features $m = \Omega\left(|D|^{\frac{2r}{1+r}}\right)$, we have

$$\left\|\bar{f}^M_{m,D,\lambda} - f_\rho\right\|_K = \mathcal{O}\left(\left(p^{\frac{1}{2}}|D|^{-\frac{r}{2(1+r)}}\right)^{M+2}\right), \tag{40}$$

where $M = l$. We complete this proof. □

## D  PROPOSITIONS

**Proposition 1** ((Liu et al., 2021)). *Let $\zeta_1, \ldots, \zeta_n$ with $n \geq 1$, be i.i.d random vectors on a separable Hilbert spaces $\mathcal{H}$ such that $H = \mathbb{E}\zeta \otimes \zeta$ is a trace class, and for any $\lambda$ there exists $\mathcal{N}_\infty(\lambda) < \infty$ such that $\langle \zeta, (H + \frac{\lambda}{2}I)^{-1}\zeta \rangle \leq \mathcal{N}_\infty(\lambda)$. Denote $H_n$ as $\frac{1}{n}\sum_{i=1}^n \zeta_i \otimes \zeta_i$. Then for any $\delta \geq 0$, with probability at least $1 - 2\delta$, the following holds*

$$\left\| (H + \frac{\lambda}{2}I)^{-1/2}(H - H_n)(H + \frac{\lambda}{2}I)^{-1/2} \right\| \leq \frac{2\log^2(2/\delta)(\mathcal{N}_\infty(\lambda) + 1)}{n} + \sqrt{\frac{2\log(2/\delta)(\mathcal{N}_\infty(\lambda) + 1)}{n}}.$$

**Proposition 2** ((Blanchard & Krämer, 2010)). *For any self-adjoint and positive semidefinite operators $A$ and $B$, if there exists $\eta > 0$ such that the following inequality holds*

$$\left\| (A + \frac{\lambda}{2}I)^{-1/2}(B - A)(A + \frac{\lambda}{2}I)^{-1/2} \right\| \leq 1 - \eta,$$

*then*

$$\left\| (A + \frac{\lambda}{2}I)^{1/2}(B + \frac{\lambda}{2}I)^{-1/2} \right\| \leq \frac{1}{\sqrt{\eta}}.$$

**Proposition 3** (Proposition 10 in Rudi & Rosasco (2017)). *For any $\delta \in (0,1], m \geq \Omega\left(2\tau^2\lambda^{-1}\log\frac{1}{\lambda\delta}\right)$ then with probability at least $1 - \delta$,*

$$|\mathcal{N}_m(\lambda) - \mathcal{N}(\lambda)| \leq 1.55\mathcal{N}(\lambda),$$

*where $\mathcal{N}_m(\lambda) = \text{Tr}\left(\left(L_m + \frac{\lambda}{2}I\right)^{-1}L_m\right)$.*

**Proposition 4** (Eq.(9) in Chen et al. (2021), Chen (2012)). *Assume that $L_K^{-r}f_\rho \in \mathcal{H}_K$ with $0 < r \leq 1$, where $L_K^r$ is the $r$-th power of $L_K$, we have $\|f_\lambda - f_\rho\|_K = \mathcal{O}(\lambda^r)$.*

Here we prove the gradient of the empirical risk of $\frac{1}{|D_j|^2}\sum \left(y_i - y_k - (\mathbf{g}^T\phi_m(\mathbf{x}_i) - \mathbf{g}^T\phi_m(\mathbf{x}_k))\right)^2 + \lambda\|\mathbf{g}\|^2$ on $\mathbf{g}$ is $4G_{m,D_j,\lambda}(\mathbf{g})$ for all $(\mathbf{x}_i, y_i), (\mathbf{x}_k, y_k) \in D_j$.

*Proof.* We have

$$\frac{\partial \frac{1}{|D_j|^2}\sum \left(y_i - y_k - (\mathbf{g}^T\phi_m(\mathbf{x}_i) - \mathbf{g}^T\phi_m(\mathbf{x}_k))\right)^2 + \lambda\|\mathbf{g}\|^2}{\partial \mathbf{g}}$$

$$= \frac{4}{|D_j|^2}\sum \left(y_i\phi_m(\mathbf{x}_k) - y_i\phi_m(\mathbf{x}_i) + \mathbf{g}^T\phi_m(\mathbf{x}_i)\phi_m(\mathbf{x}_i) - \mathbf{g}^T\phi_m(\mathbf{x}_i)\phi_m(\mathbf{x}_k)\right) + 2\lambda\mathbf{g}$$

$$= 4\left((\mathbf{\Phi}_{m,D_j}\mathbf{W}_{D_j}\mathbf{\Phi}_{m,D_j}^T + \frac{\lambda}{2}\mathbf{I})\mathbf{g} - \mathbf{\Phi}_{m,D_j}\mathbf{W}_{D_j}\bar{\mathbf{y}}_{D_j}\right).$$

So, we have the results. □

## E  SUPPLEMENTARY EXPERIMENTS

We add the experiments on the dataset Jester Joke. Jester Joke is publicly available from the following URL: http://www.grouplens.org/taxonomy/term/14 and contains over $4.1$ million continuous anonymous ratings ($-10.00$ to $+ 10.00$) of 100 jokes from 73,421 users. We group the reviewers according to the number of jokes they have reviewed. The grouping is 40-60 jokes. For a given test reviewer, 300 reference reviewers are chosen at random from the group and their rating are used to form the input vectors. 70 percent of the test reviewer's joke ratings are used for training and the rest for testing. Missing review values in the input features are populated with the median review score of the given reference reviewer. Here, we add the comparison with MPRank algorithm (Cortes et al., 2007). It is not a distributed algorithm related to this paper, but it is a representative algorithm in the field of least square ranking, so it is compared here.

Table 3: Comparison of the average testing error (standard deviation) and training time (in seconds) on Jester Joke dataset, with partitions $p = 2$ and 4 and random features $m = 30$ and 50. 2, 8, and 16 are the number of communications.

| Algorithm (m=30) | p=2 | | p=4 | |
|---|---|---|---|---|
| | Error | Time | Error | Time |
| LSRank | $0.411 \pm 0.002$ | 0.301 | $0.411 \pm 0.002$ | 0.301 |
| MPRank | $0.418 \pm 0.006$ | 0.285 | $0.418 \pm 0.006$ | 0.285 |
| DRank | $0.419 \pm 0.002$ | 0.194 | $0.421 \pm 0.003$ | 0.105 |
| DRank-C #2 | $0.415 \pm 0.002$ | 0.211 | $0.418 \pm 0.002$ | 0.155 |
| DRank-C #8 | $0.414 \pm 0.001$ | 0.252 | $0.415 \pm 0.005$ | 0.198 |
| DRank-RF | $0.420 \pm 0.001$ | 0.022 | $0.421 \pm 0.002$ | 0.010 |
| DRank-RF-C #2 | $0.417 \pm 0.002$ | 0.027 | $0.419 \pm 0.007$ | 0.014 |
| DRank-RF-C #8 | $0.415 \pm 0.003$ | 0.031 | $0.416 \pm 0.002$ | 0.017 |
| DRank-RF-C #16 | $0.413 \pm 0.003$ | 0.040 | $0.415 \pm 0.004$ | 0.021 |
| Algorithm (m=50) | p=2 | | p=4 | |
| | Error | Time | Error | Time |
| LSRank | $0.411 \pm 0.002$ | 0.301 | $0.411 \pm 0.002$ | 0.301 |
| MPRank | $0.418 \pm 0.006$ | 0.285 | $0.418 \pm 0.006$ | 0.285 |
| DRank | $0.419 \pm 0.002$ | 0.194 | $0.421 \pm 0.003$ | 0.105 |
| DRank-C #2 | $0.415 \pm 0.002$ | 0.211 | $0.418 \pm 0.002$ | 0.155 |
| DRank-C #8 | $0.414 \pm 0.001$ | 0.252 | $0.415 \pm 0.005$ | 0.198 |
| DRank-RF | $0.419 \pm 0.002$ | 0.025 | $0.420 \pm 0.001$ | 0.013 |
| DRank-RF-C #2 | $0.416 \pm 0.004$ | 0.029 | $0.418 \pm 0.001$ | 0.016 |
| DRank-RF-C #8 | $0.414 \pm 0.001$ | 0.034 | $0.415 \pm 0.002$ | 0.020 |
| DRank-RF-C #16 | $0.413 \pm 0.002$ | 0.047 | $0.414 \pm 0.002$ | 0.026 |

Table 4: Comparison of the average testing error and training time (in seconds) on simulated and real datasets under the same conditions as (Chen et al., 2021).

| Algorithm | Simulated Data | | Real Data | |
|---|---|---|---|---|
| | Error | Time | Error | Time |
| LSRank | 0.0206 | 2.5643 | 0.4902 | 4.0127 |
| DRank | 0.0216 | 0.0089 | 0.4913 | 0.0179 |
| DRank-C #8 | 0.0206 | 0.0213 | 0.4910 | 0.0454 |
| DRank-RF | 0.0217 | 0.0003 | 0.4914 | 0.0021 |
| DRank-RF-C #8 | 0.0207 | 0.0021 | 0.4910 | 0.0087 |

The empirical evaluations are given in Table 3 where the number of random features is $m = 30$ and 50 and the number of partitions is $p = 2$ and 4. In Table 3, we can find that the experimental results are similar to those on the simulated data and MovieLens dataset. The average testing errors of our methods, the exact method, MPRank, and DRank remain at the same level, which verify the effectiveness of our methods on the real dataset. The testing error of DRank-RF-C decreases with the increase of the number of communications, which demonstrates the effectiveness of the communication strategy on the real dataset. The proposed DRank-RF and Drank-RF-C have significant advantages over LSRank, MPRank, DRank, and DRank-C in the training time. These are consistent with the theoretical analysis.

We add the experiments under the same experiments setting as (Chen et al., 2021) on the datasets mentioned in the main paper. Table 4 shows the experimental results with partitions $p = 60$, dimension $q = 3$, and random features $m = 150$ on simulated dataset with the same data generating distribution as (Chen et al., 2021), and $p = 60$ and $m = 150$ on MovieLens dataset. Our algorithm DRank-RF has a significant advantage over DRank and LSRank in the training time. Under the same conditions, the testing errors of the proposed DRank-RF and DRank-RF-C are similar to those of DRank and DRank-C.

