# OpenReview forum: "Distributed Least Square Ranking with Random Features"
_ICLR.cc/2023/Conference — Submitted to ICLR 2023_

### Official Review · Reviewer_9DzG · 2022-10-25

**Confidence:** 2
**Clarity, Quality, Novelty And Reproducibility:** 1. They claim to be the first to appl…
**Correctness:** 4
**Technical Novelty And Significance:** 4
**Empirical Novelty And Significance:** 3
**Recommendation:** 8

**Strength And Weaknesses:**

Strength
The theoretical bound they derived for DRank-RF is sharper than existing state-of-the-art distributed pairwise ranking kernel learning.


**Summary Of The Paper:**

1. They study the convergence of pairwise ranking using distributed learning and random features (DRank-RF) via both theoretical assessments and numerical experiments.
2. They claim that prior work in distributed learning mainly focuses on pointwise kernel learning, and the existing work for pointwise kennel learning suffers from high computational requirements. This work explores the efficient way of doing distributed pairwise ranking kernel learning by combining distributed learning and random features for pairwise ranking kernel learning.
3. Both theoretical analysis and extensive experiments are conducted. They derive the convergence rate of the proposed method, which is sharper than that of the existing state-of-the-art distributed pairwise ranking kernel learning.
4. They propose a communication strategy to further improve the performance of DRank-RF, called DRank-RF-C.


**Summary Of The Review:**

I did not fully verify all the proof in the supple. Given that is correct, I don’t observe errors in the theoretical analysis.

---

### Official Review · Reviewer_NK1r · 2022-10-30

**Confidence:** 3
**Correctness:** 2
**Technical Novelty And Significance:** 2
**Empirical Novelty And Significance:** 3
**Recommendation:** 3

**Clarity, Quality, Novelty And Reproducibility:**

Clarity:
The description is clear.

---

Quality and Novelty:
I think that this paper has no sufficient quality and novelty for ICLR.

---

Reproducibility:
It seems that the experiment can be reproduced from the description of the paper, but it would be more helpful if you could provide the program code used in the experiment.
[6*] is also my concern.

**Strength And Weaknesses:**

I place special emphasis on comments with the mark *.

---

Strength:

[1] This is the first study to apply RF to DLSRank and DLSRank-C by (Chen et al., 2021).

---

Weakness:

[2*] I think that this study should be viewed as an RF-version of (Chen et al., 2021).
The authors should clearly state this perspective in Section 1.
For example, communication strategy has also proposed in (Chen et al., 2021), but it sounds like the authors considered the strategy newly in the current style of writing.
I cannot find a sufficient (for ICLR) novelty from (Chen et al., 2021).

[3] Some results of Table 2 overlap.
It can be simplified.

[4] I do not know the notation manner of $2^{[-2:0:5:5]}$ and $2^{[-13:2:-3]}$.
Please define it.

---

Question:

Regarding the following points, it may be that the authors' description is appropriate and there is no problem, just because I have not understood it correctly.
I do not reflect these points in my current recommendation score.
Depending on the authors' response, I may change my recommendation score.

[5] Almost all of the theorems in this paper are similar to those of (Chen et al., 2021).
Are there any difficulties in theoretical analysis associated with application of RF?

As noted in Remark 1, the authors provide a sharper convergence rate than in that of (Chen et al., 2021).
It would be better to clearly describe the reason of this improvement (is it a miss of (Chen et al., 2021) or merit of RF?).

[6*] I cannot prove the statement "Note that, the gradient ~ all $(x_i,y_i), (x_k,y_k)\in D$." located between (6) and (7).
Is this statement correct?
Pairwise learning and distributed learning may have bad compatibility:
In distributed learning, one cannot calculate the cross term between an object depending on $(x_i,y_i)\in D_1$ and an object depending on $(x_j,y_j)\in D_2$ of original pairwise objective.
I think that the statement is incorrect.

[7] Why did the authors change the experimental setting (e.g., data generating distribution in Section 6.2, and $p$ in Section 6.3) from (Chen et al., 2021)?
If the proposed methods did not work well with the previous settings, it would be better to conduct experiments with the previous settings and the changed settings, and consider and describe the reasons why the proposed methods work or do not work well.

[8*] Even for a random prediction $f$, $R(f)$ is 0.5.
I am not satisfied with all the experimental results in Table 2 in Section 6.3.
I fear that the LSRank (and DLSRank) on which this study is based may not be good.
Thus, I want the authors to additionally try other type methods such as OLS (and ordinal regression or standard classification methods since $\{y_i\}$ of MovieLens are discrete).

Results of this paper are similar to those of (Chen et al., 2021), but I think that the experiment of (Chen et al., 2021) also may not be sufficient (see [6*]).

[9*] This comment relates to [8*].
What is the advantage of LSRank compared with ordinary least squares (OLS) regression?
I am interested in the relationship between LSRank and OLS since their optimal predictors are the same conditional mean.
I would like the authors to make an experimental comparison at least.
Additionally, if there are advantages, it would be better to mention them in the paper.
I think distributed OLS is more easy.

**Summary Of The Paper:**

This study proposes variants of DLSRank and DLSRank-C by (Chen et al., 2021) that apply random features (RF).
In the experiment, RF speeded them up without reducing prediction performance in the pairwise comparison task.

**Summary Of The Review:**

I see this study as an RF-version of (Chen et al., 2021).
I cannot find a sufficient (for ICLR) novelty from (Chen et al., 2021) (see [2*]).
Also, I think that it may have critical mistakes (see [6*]).

---

### Official Review · Reviewer_R27i · 2022-10-31

**Confidence:** 3
**Clarity, Quality, Novelty And Reproducibility:** The paper is clearly written, novel, …
**Correctness:** 4
**Technical Novelty And Significance:** 3
**Empirical Novelty And Significance:** 3
**Recommendation:** 6

**Strength And Weaknesses:**

The paper is rigorous, but several results are based on the prior work of DX Zhou and collaborators.

**Summary Of The Paper:**

The authors study the statistical properties of pairwise ranking using distributed learning and random features ( DRank-RF) and establish its convergence analysis in probability. Numerical results confirm the practical aspects of the theory.

**Summary Of The Review:**

The authors develop the DRank-RF approach and study its theoretical properties. Empirical results confirm its practical utility.

---

### Comment · Reviewer_NK1r · 2022-11-24
**2nd Official Review of Paper6498 by Reviewer NK1r**

Additional question

[10] Theorem 2:
I think that $\mathcal{O}((p^{1/2}|D|^{-\frac{r}{2(1+r)}})^{M+2})$ may exceed the min-max rate (of OLS with no communication).
For example, with fixed $p$ and $(r,M)=(1,6)$, the rate is $\mathcal{O}(|D|^{-2})$.
Is this contradiction to the mini-max analysis?

---

Please give me more time to response your reply for the 1st review comment.
Sorry.

---

### Comment · Reviewer_NK1r · 2022-11-26
**5th Official Review of Paper6498 by Reviewer NK1r**

Regarding [10]

I understand that you claim that the difference between $M+1$ and $M+2$ is your improvement.

---

Additional question

[11]
I reviewed the proof regarding your above claim (difference difference between $M+1$ and $M+2$).

In the second inequality of (19), I think that you use the equation $(\Phi_\{m,D}W_\{D}\Phi_\{m+D}^T+\frac{\lambda}{2})^{-1}\Phi_\{m,D}W_\{D}\bar{y}_\{D}=\sum_\{j=1}^p\frac{|D_\{j}|^2}{\sum_\{k=1}^p|D_\{k}|^2}(\Phi_\{m,D}W_\{D}\Phi_\{m+D}^T+\frac{\lambda}{2})^{-1}\Phi_\{m,D_\{j}}W_\{D_\{j}}\bar{y}_\{D_\{j}}$ (or $\Phi_\{m,D}W_\{D}\bar{y}_\{D}=\sum_\{j=1}^p\frac{|D_\{j}|^2}{\sum_\{k=1}^p|D_\{k}|^2}\Phi_\{m,D_\{j}}W_\{D_\{j}}\bar{y}_\{D_\{j}}$).

However, I cannot prove this equation.
Please let me know how to prove this equation.
If the paper already describes this equation, please let me know where it is.

---

### Decision · Program_Chairs · 2023-01-20

**Decision:**

Reject

**Justification For Why Not Higher Score:**

I would like to thank the authors for their feedback to the reviewers' questions and concerns, which contributed highly to improving the understanding of the authors' work. Nevertheless, since there was a major update in the manuscript during the rebuttal phase and there are still remaining concerns regarding the proof details, I think the paper can benefit from a thorough revision before the authors can sell their ideas clearly, cleanly, and correctly to the community. For this reason, I decided not to recommend the acceptance of this work.

**Justification For Why Not Lower Score:**

N/A

**Metareview: Summary, Strengths And Weaknesses:**

Summary:
This paper investigate the problem of pairwise ranking based on distributed learning and random features. The authors proposed a new approach DRank-RF and studied its usefulness theoretically and experimentally.

Strength:
The proposed method is supported both theoretically and experimentally.

Weakness:
Clarity of the theoretical contribution needs improvement.